# Out-of-distribution Generalization for Total Variation based Invariant Risk Minimization

**Yuanchao Wang**[1,2]**, Zhao-Rong Lai**[1*]**, Tianqi Zhong**[3]

[1]Guangdong Key Laboratory of Data Security and Privacy Preserving,
  College of Cyber Security, Jinan University
[2]Pratt School of Engineering, Duke University
[3]Sino-French Engineer School, Beihang University
`yuanchao.wang@duke.edu, laizhr@jnu.edu.cn, zhongtianqi@buaa.edu.cn`

## ABSTRACT

Invariant risk minimization is an important general machine learning framework that has recently been interpreted as a total variation model (IRM-TV). However, how to improve out-of-distribution (OOD) generalization in the IRM-TV setting remains unsolved. In this paper, we extend IRM-TV to a Lagrangian multiplier model named OOD-TV-IRM. We find that the autonomous TV penalty hyperparameter is exactly the Lagrangian multiplier. Thus OOD-TV-IRM is essentially a primal-dual optimization model, where the primal optimization minimizes the entire invariant risk and the dual optimization strengthens the TV penalty. The objective is to reach a semi-Nash equilibrium where the balance between the training loss and OOD generalization is maintained. We also develop a convergent primal-dual algorithm that facilitates an adversarial learning scheme. Experimental results show that OOD-TV-IRM outperforms IRM-TV in most situations.

## 1 INTRODUCTION

Traditional risk minimization methods such as Empirical Risk Minimization (ERM) are widely used in machine learning. ERM generally assumes that both training and test data come from the same distribution. Based on this assumption, ERM learns model parameters by minimizing the average loss on the training data. However, the distributions between training and test data or even within the training or test data are often different in practical situations, which affects the model's performance in new environments. Besides, ERM tends to exploit correlations in the training data, even if these correlations do not hold in different environments. This may cause the model to learn some spurious features that are irrelevant to the target, so as to perform poorly in new environments. Because ERM only focuses on the average performance on a given data distribution, the model may overfit to specific training data and thus perform unstably when facing distribution changes or outliers. These problems lead to the poor generalization ability of ERM across different environments (Recht et al., 2019; Arjovsky et al., 2019; Lin et al., 2022).

The key to solving the above problems is to distinguish between invariant features and spurious features that cause distribution shifts, so that a trained model can be generalized to an unseen domain. This leads to the concept of out-of-distribution (OOD) generalization (Ben-Tal et al., 2009; Ye et al., 2022; Huang et al., 2023). One important methodology is the Invariant Risk Minimization (IRM, Arjovsky et al. 2019) criterion. It aims to extract invariant features across different environments to improve the generalization and robustness of the model (Yang et al., 2023; Xin et al., 2023). Specifically, it introduces a gradient norm penalty that measures the optimality of the virtual classifier in each environment. Then it minimizes the risk in all potential environments to ensure that the model can still work effectively when facing distribution drift. In summary, IRM copes with distribution changes by introducing environmental invariance constraints in order to make the trained model more robust in a wider range of application scenarios (Ahuja et al., 2020). There are various variants and improvements of IRM, such as Heterogeneous Risk Minimization (HRM, Liu et al. 2021), Risk Extrapolation (REx, Krueger et al. 2021), SparseIRM (Zhou et al., 2022), jointly

---

[*]Corresponding author.

learning with auxiliary information (ZIN, Lin et al. 2022), and invariant feature learning through independent variables (TIVA, Tan et al. 2023a). Diversifying spurious features (Lin et al., 2024) can also be an effective approach to improve IRM. These approaches not only provide new IRM settings or new application scenarios, but also improve the robustness and extensibility of IRM. They contribute to the theoretical understanding of OOD generalization.

A recent work reveals that the mathematical essence of IRM is a total variation (TV) model (Lai & Wang, 2024). TV measures the locally varying nature of a function, which is widely applied to different areas of mathematics and engineering, such as signal processing and image restoration. Its core idea is to eliminate noise by minimizing the TV of a function while preserving sharp discontinuities in the function (e.g., edges in an image (Dey et al., 2006)). The interpretation of IRM as a TV model not only provides a unified mathematical framework, but also reveals why the IRM approach can work effectively across different environments. The original IRM actually contains a TV term based on the $\ell_2$ norm (TV-$\ell_2$). Compared with TV-$\ell_2$, TV-$\ell_1$ further has the coarea formula that provides a geometric nature of preserving sharp discontinuities. This property is well-suited for the invariant feature extraction of IRM. Hence an IRM-TV-$\ell_1$ framework is proposed and it shows better performance than the IRM-TV-$\ell_2$ framework. When considering the learning risk as part of the full-variance model, IRM-TV-$\ell_1$ performs better in OOD generalization. This finding helps to address distributional drift and improve model robustness in machine learning.

However, IRM-TV-$\ell_1$ may not achieve complete OOD generalization, due to insufficient diversity in the training environments and the inflexible TV penalty. Lai & Wang (2024) investigate some additional requirements for IRM-TV-$\ell_1$ to achieve OOD generalization. A key point is to let the penalty parameter vary according to the invariant feature extractor; however, the authors do not specify a tractable implementation. It leaves unsolved how to construct and optimize this framework for broader practical applications, resulting in an urgent need for effective solutions.

To fill this gap, we have found that the TV penalty hyperparameter can be used as a Lagrangian multiplier. We propose to extend the IRM-TV framework to a **Lagrangian multiplier framework**, named **OOD-TV-IRM**. It is essentially a primal-dual optimization framework. The primal optimization reduces the entire invariant risk, in order to learn invariant features, while the dual optimization strengthens the TV penalty, in order to provide an adversarial interference with spurious features. The objective is to reach a **semi-Nash equilibrium** where the balance between the training loss and OOD generalization is maintained. We also develop a **convergent primal-dual algorithm** that facilitates an **adversarial learning** scheme. This work not only extends the theoretical foundation of previous work, but also provides a concrete implementation scheme to improve OOD generalization of machine learning methods across unknown environments.

## 2    PRELIMINARIES AND RELATED WORKS

### 2.1    INVARIANT RISK MINIMIZATION

In machine learning tasks, we often work with a data set consisting of multiple samples, where each sample includes input and output variables. These samples are typically drawn from various environments, but in many scenarios, the specific environment labels are not explicitly available. As a result, the challenge is to train a model that can generalize well across these different environments, even when the environment information is unknown.

IRM (Arjovsky et al., 2019) addresses this challenge by structuring the model into two components: a feature extractor and a classifier. The feature extractor is responsible for identifying invariant features from the input data, while the classifier makes predictions based on these features. IRM aims to minimize the invariant risk across different training environments, in order to find a representation that performs consistently well across all environments.

For a given training data set of $n$ samples $\mathcal{D} := \{(x_i, y_i) \in \mathcal{X} \times \mathcal{Y}\}_{i=1}^n$, the empirical risk in a given environment $e$ is computed using a loss function $\mathcal{L}$, which measures the discrepancy between the predicted value and the true output. This risk is expressed as:

$$R(w \circ \Phi, e) := \frac{1}{n} \sum_{i=1}^n \mathcal{L}(w \circ \Phi(x_i), y_i, e), \tag{1}$$

which represents the mean loss over $n$ samples in environment $e$. IRM simultaneously learns an invariant feature extractor $\Phi$ and a classifier $w$ across multiple training environments, so that the risk is minimized in all environments:

$$\min_{w,\Phi} \sum_{e \in \mathcal{E}_{\text{tr}}} R(w \circ \Phi, e), \quad \text{s. t.} \quad w \in \arg\min_{w'} R(w' \circ \Phi, e), \forall e \in \mathcal{E}_{\text{tr}}, \tag{2}$$

where the classifier $w$ should be the optimal solution for minimizing the risk in each environment $e$. However, it is a challenging bilevel optimization problem. Hence Arjovsky et al. (2019) further propose a practical variant of IRM as follows:

$$\min_{\Phi} \sum_{e \in \mathcal{E}_{\text{tr}}} \left\{ R(1 \circ \Phi, e) + \lambda \|\nabla_w|_{w=1} R(w \circ \Phi, e)\|_2^2 \right\}. \tag{3}$$

In this version, the classifier $w$ is set to a scalar value of 1, and a gradient norm penalty term is introduced to convert the objective into a single-level optimization problem.

ZIN (Lin et al., 2022) further extends the IRM framework by introducing auxiliary information $z_i \in \mathcal{Z}$ to simultaneously learn environment partitioning and invariant feature representation. It assumes that the space of training environments $\mathcal{E}_{\text{tr}}$ is the convex hull of $E$ linearly independent fundamental environments. By learning a mapping $\rho : \mathcal{Z} \to \Delta_E$ that provides weights for different environments, ZIN optimizes the following objective:

$$\min_{w,\Phi} \left\{ R\left(w \circ \Phi, \frac{1}{E} 1_{(E)}\right) \cdot 1_{(E)} + \lambda \max_{\rho} \|\nabla_w R(w \circ \Phi, \rho)\|_2^2 \right\}. \tag{4}$$

By this means, ZIN can effectively learn invariant features even without explicit environment partition information.

## 2.2 TOTAL VARIATION

TV is a mathematical operator widely recognized for its ability to measure the global variability of a function. It has been extensively applied in various fields such as optimal control, data transmission, and signal processing. The central concept behind TV is to quantify the overall amount of change in a function across its domain. It is able to preserve sharp discontinuities (such as edges in images) while removing noise, making it a useful tool in image restoration and signal denoising. This property is further elucidated through the coarea formula (Chen et al., 2006). For a function $f \in L^1(\Omega)$, where $\Omega$ is an open subset of $\mathbb{R}^d$,

$$\int_{\Omega} |\nabla f| := \int_{-\infty}^{\infty} \int_{f^{-1}(\gamma)} ds \, d\gamma, \tag{5}$$

where $f^{-1}(\gamma)$ represents the level set of $f$ at the value $\gamma$. This formulation shows that TV integrates over all the contours of the function, reinforcing its capability in capturing piecewise-constant features. Mumford & Shah (1985) and Rudin et al. (1992) propose the following TV-$\ell_2$ and TV-$\ell_1$ models, respectively.

$$\inf_{f \in L^2(\Omega)} \left\{ \int_{\Omega} |\nabla f|^2 + \lambda \int_{\Omega} (f - \tilde{f})^2 \, dx \right\}, \quad \inf_{f \in L^2(\Omega)} \left\{ \int_{\Omega} |\nabla f| + \lambda \int_{\Omega} (f - \tilde{f})^2 \, dx \right\}, \tag{6}$$

where $\tilde{f} \in L^2(\Omega)$ is the ground-truth signal and $\lambda$ is a hyperparameter that affects accuracy. These models aim to preserve target features in the approximation $f$ and leave noise in the residual $(f - \tilde{f})$. In general, TV-$\ell_1$ can better preserve useful features and sharp discontinuities in the approximation $f$.

## 2.3 INVARIANT RISK MINIMIZATION BASED ON TOTAL VARIATION

It can be seen that the penalties of (3) and (4) are similar to a TV-$\ell_2$ term, which has been verified by Lai & Wang (2024) under some mild conditions, such as measurability of the involved functions, non-correlation between the feature extractor and the environment variable, representability of $w$ by $e$, etc. In this sense, (3) and (4) are actually IRM-TV-$\ell_2$ and Minimax-TV-$\ell_2$ models. Considering

the superiority of TV-$\ell_1$ over TV-$\ell_2$ in sharp feature preservation, Lai & Wang (2024) further propose the following IRM-TV-$\ell_1$ (7) and Minimax-TV-$\ell_1$ (8) models:

$$\min_{\Phi} \ \left\{ \mathbb{E}_w[R(w \circ \Phi)] + \lambda(\mathbb{E}_w[|\nabla_w R(w \circ \Phi)|])^2 \right\}, \tag{7}$$

$$\min_{\Phi} \ \left\{ \mathbb{E}_{w \leftarrow \frac{1_{(E)}}{E}}[R(w \circ \Phi)] + \lambda \max_{\rho}(\mathbb{E}_{w \leftarrow \rho}[|\nabla_w R(w \circ \Phi)|])^2 \right\}, \tag{8}$$

where $\mathbb{E}_{w \leftarrow \rho}$ denotes the mathematical expectation with respect to (w.r.t.) $w$ whose measure is induced by $\rho$, and $\frac{1_{(E)}}{E}$ denotes the uniform probability measure for the environment $e$. The maximization w.r.t. $\rho$ identifies the worst-case inferred environment causing the greatest variation in the risk function. Hence, Minimax-TV-$\ell_1$ can be seen as the TV-$\ell_1$ version of ZIN. The corresponding coarea formula for IRM-TV-$\ell_1$ or Minimax-TV-$\ell_1$ is

$$\int_{\Omega} |\nabla_w R(w \circ \Phi)| \, d\nu = \int_{-\infty}^{\infty} \int_{\{w \in \Omega : R(w \circ \Phi) = \gamma\}} ds \, d\gamma, \tag{9}$$

where $\{w \in \Omega : R(w \circ \Phi) = \gamma\}$ denotes the level set, and $s$ is the Hausdorff measure in the corresponding dimensions. This formula ensures that IRM-TV-$\ell_1$ or Minimax-TV-$\ell_1$ preserves essential structural features while maintaining robustness across different environments. In the rest of this paper, we abbreviate IRM-TV-$\ell_1$ as IRM-TV if not specified otherwise.

## 3 METHODOLOGY

### 3.1 AUTONOMOUS TOTAL VARIATION PENALTY AND LAGRANGIAN MULTIPLIER

OOD generalization represents the generalization ability of a trained model to an unseen domain. From the perspective of IRM-TV, it can be represented by (Ben-Tal et al., 2009; Arjovsky et al., 2019; Lin et al., 2022; Lai & Wang, 2024)

$$\min_{\Phi} \max_{w} R(w \circ \Phi). \tag{10}$$

In general, IRM-TV cannot achieve OOD generalization without additional conditions, due to insufficient diversity in the training environments and the inflexible TV penalty. However, Lai & Wang (2024) indicate that the penalty hyperparameter $\lambda_{\Phi}$ of IRM-TV should be variable according to the invariant feature extractor $\Phi$:

$$\min_{\Phi} \ \left\{ \mathbb{E}_w[R(w \circ \Phi)] + \lambda_{\Phi}(\mathbb{E}_w[|\nabla_w R(w \circ \Phi)|])^2 \right\}, \tag{11}$$

$$\min_{\Phi} \ \left\{ \mathbb{E}_{w \leftarrow \frac{1_{(E)}}{E}}[R(w \circ \Phi)] + \lambda_{\Phi} \max_{\rho}(\mathbb{E}_{w \leftarrow \rho}[|\nabla_w R(w \circ \Phi)|])^2 \right\}. \tag{12}$$

The main reason is that a variable $\lambda_{\Phi}$ can fill the gap between the objective value of (11) or (12) and the objective value of (10). Lai & Wang (2024) further indicate that this variable $\lambda_{\Phi}$ exists in general situations. However, they do not specify any tractable and concrete implementation of $\lambda_{\Phi}$.

To fill this gap, we find that $\lambda_{\Phi}$ actually serves as a Lagrangian multiplier in (11) and (12). Hence we use it as an autonomous parameter, which is also taken into the training scheme. Specifically, the parameters for the invariant feature extractor, also denoted by $\Phi$, are directly taken as inputs for $\lambda$. Next, $\lambda$ can be parameterized by another set of parameters $\Psi$. By this means, $\lambda$ can be represented as a function of both $\Psi$ and $\Phi$: $\lambda(\Psi, \Phi)$. Then $\lambda$ not only depends on $\Phi$, but also adjusts its strength through $\Psi$. Now it can be deployed in a primal-dual optimization model, as illustrated in the next section.

### 3.2 PRIMAL-DUAL OPTIMIZATION AND SEMI-NASH EQUILIBRIUM

Replacing $\lambda_{\Phi}$ by $\lambda(\Psi, \Phi)$ in (11) and (12), we obtain the corresponding Lagrangian functions:

$$g(\Psi, \Phi) := \mathbb{E}_w[R(w \circ \Phi)] + \lambda(\Psi, \Phi)(\mathbb{E}_w[|\nabla_w R(w \circ \Phi)|])^2, \tag{13}$$

$$h(\rho, \Psi, \Phi) := \mathbb{E}_{w \leftarrow \frac{1_{(E)}}{E}}[R(w \circ \Phi)] + \lambda(\Psi, \Phi)(\mathbb{E}_{w \leftarrow \rho}[|\nabla_w R(w \circ \Phi)|])^2. \tag{14}$$

Applying the OOD generalization criterion (10) in the literature of IRM, we adopt a minimax scheme to optimize $\Phi$ and $\Psi$.

- **Outer Minimization (Primal):** the primal variable $\Phi$ is trained to minimize the entire invariant risk, in order to capture invariant features across different environments.
- **Inner Maximization (Dual):** the dual variable $\Psi$ (or $(\rho, \Psi)$) is trained to maximize the autonomous TV penalty $\lambda(\Psi, \Phi)(\mathbb{E}_w[|\nabla_w R(w \circ \Phi)|])^2$ (or $\lambda(\Psi, \Phi)(\mathbb{E}_{w \leftarrow \rho}[|\nabla_w R(w \circ \Phi)|])^2$), which confronts the most adverse scenarios with spurious features.

They lead to the following proposed OOD-TV-IRM (15) and OOD-TV-Minimax (16) models:

$$\min_\Phi \max_\Psi g(\Psi, \Phi) := \min_\Phi \left\{ \mathbb{E}_w[R(w \circ \Phi)] + \max_\Psi \left[ \lambda(\Psi, \Phi)(\mathbb{E}_w[|\nabla_w R(w \circ \Phi)|])^2 \right] \right\}, \quad (15)$$

$$\min_\Phi \max_{\rho, \Psi} h(\rho, \Psi, \Phi) := \min_\Phi \left\{ \mathbb{E}_{w \leftarrow \frac{1_{(E)}}{E}}[R(w \circ \Phi)] + \max_{\rho, \Psi} \left[ \lambda(\Psi, \Phi)(\mathbb{E}_{w \leftarrow \rho}[|\nabla_w R(w \circ \Phi)|])^2 \right] \right\}. \quad (16)$$

They are essentially primal-dual optimization models. To further understand their functions, we introduce the concept of semi-Nash equilibrium in the context of this paper, without loss of generality.

**Definition 1.** A semi-Nash equilibrium $(\Psi^*, \Phi^*)$ of a Lagrangian function $g(\Psi, \Phi)$ satisfies the following conditions:

1. $g(\Psi^*, \Phi^*) \geqslant g(\Psi, \Phi^*)$ for any $\Psi$ in the parameter space.

2. $g(\Psi^*, \Phi^*) = g(\Psi_{max}(\Phi^*), \Phi^*) \leqslant g(\Psi_{max}(\Phi), \Phi)$ for any $\Phi$ in the parameter space, where $\Psi_{max}(\Phi) \in \arg\max_\Psi \left[ \lambda(\Psi, \Phi)(\mathbb{E}_w[|\nabla_w R(w \circ \Phi)|])^2 \right]$. In other words, $\Psi_{max}(\Phi)$ is a solution to the dual optimization with $\Phi$ fixed.

A semi-Nash equilibrium $(\rho^*, \Psi^*, \Phi^*)$ of a Lagrangian function $h(\rho, \Psi, \Phi)$ follows the same definition with the unified dual variable $(\rho, \Psi)$.

**Remark:** in the context of game theory, OOD-TV-IRM (15) (or OOD-TV-Minimax Eq. 16) can be seen as a two-player zero-sum game, where $g(\Psi, \Phi)$ and $-g(\Psi, \Phi)$ are the gains for players $\Psi$ and $\Phi$, respectively. Item 1 in Definition 1 indicates that $\Psi^*$ is the best strategy for dual optimization given $\Phi^*$ (which is independent of $\Psi$). On the other hand, Item 2 indicates that $\Phi^*$ is the conditional best strategy for primal optimization only when $\Psi_{max}(\Phi)$ is optimized over each $\Phi$. In this sense, $(\Psi^*, \Phi^*)$ is only a semi-Nash equilibrium instead of a full Nash equilibrium. It accords with our task where $\Phi$ is the feature extractor and primary variable that should confront the most adverse scenarios, while $\Psi$ is the TV penalty parameter and secondary variable that provides OOD generalization depending on $\Phi$.

**Theorem 2.** *Assume that $R(w \circ \Phi)$ is continuous w.r.t. $(w, \Phi)$ and differentiable w.r.t. $w$, $\lambda(\Psi, \Phi)$ is continuous w.r.t. $(\Psi, \Phi)$, and the feasible set of parameters $(\rho, \Psi, \Phi)$ is bounded and closed. Then any solution to OOD-TV-IRM (15) or OOD-TV-Minimax (16) is a semi-Nash equilibrium.*

*Proof.* The proof is given in Appendix A.1. $\qquad\square$

Most of the widely-used loss functions and deep neural networks satisfy the conditions of Theorem 2, such as the squared loss, cross-entropy loss, multilayer perceptron, convolutional neural networks, etc. In practical computation, a bounded and closed feasible parameter set is also satisfied due to the maximum float point value. Hence Theorem 2 is applicable to general situations. It indicates that OOD-TV-IRM and OOD-TV-Minimax balance between the training loss and OOD generalization at a semi-Nash equilibrium.

### 3.3 PRIMAL-DUAL ALGORITHM AND ADVERSARIAL LEARNING

We develop a primal-dual algorithm to solve OOD-TV-IRM (15) or OOD-TV-Minimax (16). To do this, we further assume that $R(w \circ \Phi)$ and $\lambda(\Psi, \Phi)$ are differentiable w.r.t. their corresponding arguments $w$, $\Phi$, and $\Psi$. As indicated in (Lai & Wang, 2024), $|\nabla_w R(w \circ \Phi)|$ in (15) or (16) is non-differentiable w.r.t. $\Phi$. Hence we adopt the subgradient descent method to update these parameters, as illustrated in Appendix A.2. Then the primal and dual updates for (15) or (16) are:

$$\begin{cases} \Phi^{(k+1)} = \Phi^{(k)} - \eta_1^{(k)} \partial_\Phi g(\Psi^{(k)}, \Phi^{(k)}), \\ \Psi^{(k+1)} = \Psi^{(k)} + \eta_2^{(k)} \nabla_\Psi g(\Psi^{(k)}, \Phi^{(k+1)}); \end{cases} \quad (17)$$

$$\begin{cases} \Phi^{(k+1)} = \Phi^{(k)} - \eta_1^{(k)} \partial_\Phi h(\rho^{(k)}, \Psi^{(k)}, \Phi^{(k)}), \\ (\rho^{(k+1)}, \Psi^{(k+1)}) = (\rho^{(k)}, \Psi^{(k)}) + \eta_2^{(k)} \nabla_{(\rho, \Psi)} h(\rho^{(k)}, \Psi^{(k)}, \Phi^{(k+1)}), \end{cases} \quad (18)$$

where $\eta_1^{(k)}, \eta_2^{(k)} > 0$ are the learning rates for the primal and dual updates, respectively. The computation of the above (sub)gradients is illustrated in Appendix A.2. In general machine learning scenarios, the smoothness and convexity of the Lagrangian functions $g$ and $h$ are usually unknown, thus $\eta_1^{(k)}$ and $\eta_2^{(k)}$ should be set according to different specific tasks. In addition, we provide a convergent scheme as follows.

**Theorem 3.** *The same assumptions in Theorem 2 are used. By setting*

$$\eta_1^{(k)} := \begin{cases} \frac{1}{k^p \|\partial_\Phi g(\Psi^{(k)}, \Phi^{(k)})\|_2}, & \text{if} \quad \partial_\Phi g(\Psi^{(k)}, \Phi^{(k)}) \neq 0; \\ 0, & \text{if} \quad \partial_\Phi g(\Psi^{(k)}, \Phi^{(k)}) = 0. \end{cases}$$

$$\eta_2^{(k)} := \begin{cases} \frac{1}{k^p \|\nabla_\Psi g(\Psi^{(k)}, \Phi^{(k+1)})\|_2}, & \text{if} \quad \nabla_\Psi g(\Psi^{(k)}, \Phi^{(k+1)}) \neq 0; \\ 0, & \text{if} \quad \nabla_\Psi g(\Psi^{(k)}, \Phi^{(k+1)}) = 0. \end{cases} \tag{19}$$

$$or \quad \eta_1^{(k)} := \begin{cases} \frac{1}{k^p \|\partial_\Phi h(\rho^{(k)}, \Psi^{(k)}, \Phi^{(k)})\|_2}, & \text{if} \quad \partial_\Phi h(\rho^{(k)}, \Psi^{(k)}, \Phi^{(k)}) \neq 0; \\ 0, & \text{if} \quad \partial_\Phi h(\rho^{(k)}, \Psi^{(k)}, \Phi^{(k)}) = 0. \end{cases}$$

$$\eta_2^{(k)} := \begin{cases} \frac{1}{k^p \|\nabla_{(\rho,\Psi)} h(\rho^{(k)}, \Psi^{(k)}, \Phi^{(k+1)})\|_2}, & \text{if} \quad \nabla_{(\rho,\Psi)} h(\rho^{(k)}, \Psi^{(k)}, \Phi^{(k+1)}) \neq 0; \\ 0, & \text{if} \quad \nabla_{(\rho,\Psi)} h(\rho^{(k)}, \Psi^{(k)}, \Phi^{(k+1)}) = 0. \end{cases} \tag{20}$$

*with an arbitrary $p > 1$, the primal-dual algorithm (17) or (18) achieves convergent sequences $\{(\Psi^{(k)}, \Phi^{(k)})\}_{k=1}^\infty$ or $\{(\rho^{(k)}, \Psi^{(k)}, \Phi^{(k)})\}_{k=1}^\infty$, respectively. Moreover, its computational complexity is $\mathcal{O}(((p-1)\epsilon)^{-\frac{1}{p-1}})$ to achieve a convergence tolerance of $\epsilon > 0$.*

*Proof.* The proof is given in Appendix A.2. □

In fact, (17) or (18) facilitates adversarial learning to dynamically adjust model parameters $(\Psi, \Phi)$ or $(\rho, \Psi, \Phi)$ to balance training loss and OOD generalization. This approach extends the theoretical work of IRM-TV by providing a practical, dynamic implementation of $\lambda(\Psi, \Phi)$ that adapts to the most adverse environments.

**Remark:** the above OOD-TV-IRM and OOD-TV-Minimax models as well as the primal-dual algorithm can also be applied to the TV-$\ell_2$ versions, which are omitted here since they are simpler differentiable cases.

### 3.4 IMPLEMENTATION WITH NEURAL NETWORKS

First, we note that all the theoretical results in this section hold with any functions satisfying the corresponding conditions (e.g., continuity, differentiability), not limited to neural networks. To provide a specific instance, we implement $\lambda(\Psi, \Phi)$ with neural networks that take $\Psi$ and $\Phi$ as network trainable parameters and network inputs, respectively. Besides $\lambda(\Psi, \Phi)$, $\Phi$ and $\rho$ can also be developed by neural networks. Specifically, $\Phi : \mathcal{X} \to \mathcal{H}$ is the feature extractor that maps a sample $x$ from the sample space $\mathcal{X}$ to the feature space $\mathcal{H}$. It can be implemented by different types of neural networks depending on the practical situation, as shown in Table 1. The data sets correspond to different synthetic or real-world tasks, which will be illustrated in Section 4. For example, $\Phi$ can be a multilayer perceptron in Simulation, or a convolutional neural network in Landcover. Similarly, $\rho : \mathcal{Z} \to \Delta^E$ is the environment inference operator that maps the auxiliary variables $z \in \mathcal{Z}$ to the probability space $\Delta^E := \{v \in \mathbb{R}^E : v \geqslant 0_{(E)} \text{ and } v \cdot 1_{(E)} = 1\}$, where $\mathbb{R}^E$ denotes the $E$-dimensional real space (i.e., $E$ environments). Moreover, the architectures of $\rho$ are developed by multilayer perceptrons, which are compatible with the corresponding $\Phi$, as shown in Table 1.

Without ambiguity, we can also use $\Phi$ to denote the parameters of the feature extractor. Then $\lambda(\Psi, \Phi)$ can be implemented as a neural network that takes $\Phi$ as input and $\Psi$ as network parameters. We provide tractable architectures of $\lambda$ in Table 1, which should also be compatible with the corresponding $\Phi$. With the primal-dual algorithm in Section 3.3, $\Phi$ and $\Psi$ can be learned in an adversarial way to improve OOD generalization, which puts Theorem 2 into practice.

## 4 EXPERIMENTAL RESULTS

We conduct experiments on 7 simulation and real-world data sets to evaluate the effectiveness of the proposed OOD-TV-IRM and OOD-TV-Minimax in different OOD generalization tasks. These

Table 1: Network architectures of the invariant feature extractor $\Phi$, the environment inference operator $\rho$, and the TV penalty strength $\lambda$ w.r.t. their corresponding inputs.

| DATA SET | $x \to \Phi$ | $z \to \rho$ |
|---|---|---|
| SIMULATION | LINEAR(15, 1) | LINEAR(1, 16)→RELU()→LINEAR(16, 1) →SIGMOID() |
| CELEBA | LINEAR(512, 16)→RELU()→LINEAR(16, 1) | LINEAR(7, 16)→RELU()→LINEAR(16, 1)→SIGMOID() |
| LANDCOVER | CONV1D(8, 32, 5)→RELU()→CONV1D(32, 32, 3) →RELU()→MAXPOOL1D(2, 2)→CONV1D(32, 64, 3) →RELU()→MAXPOOL1D(2, 2)→CONV1D(64, 6, 3) →RELU()→AVEPOOL1D(1) | LINEAR(2, 16)→RELU()→LINEAR(16, 2)→SOFTMAX() |
| ADULT | LINEAR(59, 16)→RELU()→LINEAR(16, 1) | LINEAR(6, 16)→RELU()→LINEAR(16, 4)→SOFTMAX() |
| HOUSE PRICE | LINEAR(15,32) →RELU() → LINEAR(32,1) | LINEAR(1,64)→ RELU() → LINEAR(64,4) → SOFTMAX() |
| COLORED MNIST | CONV2D(3,32,3) → MAXPOOL2D(2,2) →CONV2D(32,64,3) → MAXPOOL2D(2,2) → CONV2D(64,128,3) → MAXPOOL2D(2,2) → LINEAR(128*3*3,128)→ LINEAR(128,10) | LINEAR(3,16) → RELU() → LINEAR(16,1) → SOFTMAX() |
| NICO | RESNET-18() | LINEAR(3,16) → RELU() → LINEAR(16,1)→ SOFTMAX() |

| DATA SET | $\Phi \to \lambda$ |
|---|---|
| SIMULATION | LINEAR(11, 1) → RELU() → LINEAR(1, 1) → SOFTPLUS() |
| CELEBA | LINEAR(8225, 16) → RELU() → LINEAR(16, 1) → SOFTPLUS() |
| LANDCOVER | LINEAR(173574, 32) → RELU() → LINEAR(32, 1) → SOFTPLUS() |
| ADULT | LINEAR(977, 16) → RELU() → LINEAR(16, 1) → SOFTPLUS() |
| HOUSE PRICE | LINEAR(545, 32) → RELU() → LINEAR(32, 16) → SOFTPLUS() → LINEAR(16,1) |
| COLORED MNIST | LINEAR(242122,32) → RELU() → LINEAR(32,1) → SOFTPLUS() |
| NICO | LINEAR(11180113,32) → RELU() → LINEAR(32,1) → SOFTPLUS() |

tasks largely follow the settings in the literature of IRM (Lin et al., 2022; Lai & Wang, 2024). We also include the original IRM (Arjovsky et al., 2019), ZIN (Lin et al., 2022), IRM-TV-$\ell_1$, Minimax-TV-$\ell_1$ (Lai & Wang, 2024), EIIL (Creager et al., 2021), LfF (Nam et al., 2020), and TIVA (Tan et al., 2023b) frameworks in comparisons to verify the effectiveness of the proposed primal-dual optimization and adversarial learning scheme. Note that IRM and ZIN are equivalent to IRM-TV-$\ell_2$ and Minimax-TV-$\ell_2$, respectively. Each experiment is repeated 10 times to record the mean and standard deviation (STD) of the results for each compared method. The code is available at https://github.com/laizhr/OOD-TV-IRM.

## 4.1 SIMULATION DATA

The simulation data consists of temporal heterogeneity observations with distributional shift w.r.t time, which is used in (Lin et al., 2022; Tan et al., 2023a) to evaluate OOD generalization. Details of generating this data set are provided in Appendix A.3. Among all the parameter settings, the most challenging one is $(p_s^-, p_s^+, p_v) = (0.999, 0.9, 0.8)$, thus we use this setting in the evaluation.

Table 2 (left) shows the mean and worst accuracies of different methods over the four test environments on simulation data. The proposed OOD-TV-based methods outperform their counterparts in all the mean and worst accuracy cases. Moreover, OOD-TV-IRM-$\ell_1$ achieves the highest accuracies among all the compared methods in all the cases, while OOD-TV-Minimax-$\ell_2$ achieves the highest accuracies among all the compared methods without environment partition in all the cases. Hence the OOD-TV-based methods achieve the best performance regardless of whether the environment partition is available or not.

Table 2: Accuracies of different methods on simulation data (left) and CelebA (right).

| Method | Mean ± STD | Worst ± STD | Method | Mean ± STD | Worst ± STD |
|---|---|---|---|---|---|
| EIIL | $0.8161 \pm 0.0364$ | $0.7159 \pm 0.0540$ | EIIL | $0.7323 \pm 0.0284$ | $0.7240 \pm 0.0291$ |
| LfF | $0.7240 \pm 0.0894$ | $0.6965 \pm 0.1049$ | LfF | $0.6092 \pm 0.0529$ | $0.5983 \pm 0.0598$ |
| TIVA | $0.8274 \pm 0.0454$ | $0.8139 \pm 0.0479$ | TIVA | $0.7695 \pm 0.0199$ | $0.7283 \pm 0.0209$ |
| ZIN | $0.7916 \pm 0.0349$ | $0.7338 \pm 0.0573$ | ZIN | $0.7155 \pm 0.0298$ | $0.6034 \pm 0.0544$ |
| OOD-TV-Minimax-$\ell_2$ | $\mathbf{0.8916 \pm 0.0202}$ | $\mathbf{0.8858 \pm 0.0223}$ | OOD-TV-Minimax-$\ell_2$ | $\mathbf{0.7608 \pm 0.0187}$ | $\mathbf{0.6895 \pm 0.0243}$ |
| Minimax-TV-$\ell_1$ | $0.7555 \pm 0.0742$ | $0.6931 \pm 0.0965$ | Minimax-TV-$\ell_1$ | $0.6876 \pm 0.0190$ | $0.5541 \pm 0.0301$ |
| OOD-TV-Minimax-$\ell_1$ | $\mathbf{0.8379 \pm 0.0418}$ | $\mathbf{0.7805 \pm 0.0612}$ | OOD-TV-Minimax-$\ell_1$ | $\mathbf{0.7044 \pm 0.0361}$ | $\mathbf{0.6180 \pm 0.0580}$ |
| IRM | $0.8670 \pm 0.0193$ | $0.8559 \pm 0.0285$ | IRM | $0.7630 \pm 0.0261$ | $0.7410 \pm 0.0215$ |
| OOD-TV-IRM-$\ell_2$ | $\mathbf{0.8904 \pm 0.0150}$ | $\mathbf{0.8831 \pm 0.0195}$ | OOD-TV-IRM-$\ell_2$ | $\mathbf{0.7786 \pm 0.0157}$ | $\mathbf{0.7549 \pm 0.0227}$ |
| IRM-TV-$\ell_1$ | $0.8713 \pm 0.0201$ | $0.8618 \pm 0.0209$ | IRM-TV-$\ell_1$ | $0.7751 \pm 0.0203$ | $0.7461 \pm 0.0170$ |
| OOD-TV-IRM-$\ell_1$ | $\mathbf{0.8944 \pm 0.0150}$ | $\mathbf{0.8877 \pm 0.0168}$ | OOD-TV-IRM-$\ell_1$ | $\mathbf{0.7945 \pm 0.0162}$ | $\mathbf{0.7560 \pm 0.0170}$ |

## 4.2 CELEBA

The CelebA data set (Liu et al., 2015) contains face images of celebrities. The task is to identify smiling faces, which are deliberately correlated with the gender variable. The 512-dimensional deep features of the face images are extracted using a pre-trained ResNet18 model (He et al., 2016), while the invariant features are learned using subsequent multilayer perceptrons (MLP). EIIL, LfF, TIVA, ZIN, OOD-TV-Minimax-$\ell_2$, Minimax-TV-$\ell_1$ and OOD-TV-Minimax-$\ell_1$ take seven additional descriptive variables for environment inference, including *Young*, *Blond Hair*, *Eyeglasses*, *High Cheekbones*, *Big Nose*, *Bags Under Eyes*, and *Chubby*. As for IRM, OOD-TV-IRM-$\ell_2$, IRM-TV-$\ell_1$, and OOD-TV-IRM-$\ell_1$, they use the gender variable as the environment indicator.

Table 2 (right) shows the accuracies of different methods on CelebA. The proposed OOD-TV-based methods outperform their counterparts in all the 8 cases with both mean and worst accuracies. Moreover, OOD-TV-IRM-$\ell_1$ achieves the highest accuracies among all the competitors. Hence OOD-TV-IRM improves OOD generalization compared with IRM-TV.

## 4.3 LANDCOVER

The Landcover data set consists of time series data and the corresponding land cover types derived from satellite images (Gislason et al., 2006; Russwurm et al., 2020; Xie et al., 2021). The input data has dimensions of $46 \times 8$ and is used to identify one of six land cover types. The invariant feature extractor $\Phi$ is implemented as a 1D-CNN to process the time series input, following the approaches of Xie et al. (2021) and Lin et al. (2022). In scenarios where ground-truth environment partitions are unavailable, latitude and longitude are used as auxiliary information for environment inference. All methods are trained on non-African data, and then tested on both non-African (from regions not overlapping with the training data) and African regions. This is a complex and challenging experiment among the 7 experiments of this paper.

Table 3 (left) shows the accuracies of different methods on Landcover. The proposed OOD-TV-based methods outperform their counterparts in all the 8 cases with both mean and worst accuracies. Hence the OOD-TV-based methods show competitive performance in this challenging task.

Table 3: Accuracies of different methods on Landcover (left) and adult income prediction (right).

| Method | Mean $\pm$ STD | Worst $\pm$ STD | Method | Mean $\pm$ STD | Worst $\pm$ STD |
|---|---|---|---|---|---|
| EIIL | $0.6653 \pm 0.0152$ | $0.6614 \pm 0.0156$ | EIIL | $0.7734 \pm 0.0120$ | $0.7395 \pm 0.0132$ |
| LfF | $0.5097 \pm 0.0567$ | $0.5017 \pm 0.0580$ | LfF | $0.7785 \pm 0.0089$ | $0.7545 \pm 0.0084$ |
| TIVA | $0.5388 \pm 0.0154$ | $0.5351 \pm 0.0175$ | TIVA | $0.8226 \pm 0.0071$ | $0.8085 \pm 0.0086$ |
| ZIN | $0.6511 \pm 0.0254$ | $0.6468 \pm 0.0228$ | ZIN | $0.8201 \pm 0.0093$ | $0.8058 \pm 0.0101$ |
| OOD-TV-Minimax-$\ell_2$ | $\mathbf{0.6530 \pm 0.0178}$ | $\mathbf{0.6490 \pm 0.0124}$ | OOD-TV-Minimax-$\ell_2$ | $\mathbf{0.8345 \pm 0.0055}$ | $\mathbf{0.8105 \pm 0.0063}$ |
| Minimax-TV-$\ell_1$ | $0.6573 \pm 0.0219$ | $0.6551 \pm 0.0176$ | Minimax-TV-$\ell_1$ | $0.8101 \pm 0.0082$ | $0.7958 \pm 0.0103$ |
| OOD-TV-Minimax-$\ell_1$ | $\mathbf{0.6604 \pm 0.0151}$ | $\mathbf{0.6566 \pm 0.0150}$ | OOD-TV-Minimax-$\ell_1$ | $\mathbf{0.8330 \pm 0.0058}$ | $\mathbf{0.8093 \pm 0.0075}$ |
| IRM | $0.6472 \pm 0.0185$ | $0.6427 \pm 0.0154$ | IRM | $0.8257 \pm 0.0069$ | $0.8052 \pm 0.0112$ |
| OOD-TV-IRM-$\ell_2$ | $\mathbf{0.6705 \pm 0.0132}$ | $\mathbf{0.6672 \pm 0.0174}$ | OOD-TV-IRM-$\ell_2$ | $\mathbf{0.8369 \pm 0.0109}$ | $\mathbf{0.8103 \pm 0.0091}$ |
| IRM-TV-$\ell_1$ | $0.6422 \pm 0.0196$ | $0.6390 \pm 0.0208$ | IRM-TV-$\ell_1$ | $0.8298 \pm 0.0074$ | $0.8054 \pm 0.0075$ |
| OOD-TV-IRM-$\ell_1$ | $\mathbf{0.6672 \pm 0.0182}$ | $\mathbf{0.6636 \pm 0.0197}$ | OOD-TV-IRM-$\ell_1$ | $\mathbf{0.8435 \pm 0.0089}$ | $\mathbf{0.8197 \pm 0.0091}$ |

## 4.4 ADULT INCOME PREDICTION

This task uses the Adult data set[1] to predict whether an individual's income exceeds $50K per year based on census data. The data set is split into four subgroups representing different environments based on *race* $\in$ {Black, Non-Black} and *sex* $\in$ {Male, Female}. Two-thirds of the data from the Black Male and Non-Black Female subgroups are randomly selected for training, and the compared methods are verified across all four subgroups using the remaining data. Six integer variables — *Age*, *FNLWGT*, *Education-Number*, *Capital-Gain*, *Capital-Loss*, and *Hours-Per-Week* — are fed into EIIL, LfF, TIVA, ZIN, OOD-TV-Minimax-$\ell_2$, Minimax-TV-$\ell_1$ and OOD-TV-Minimax-$\ell_1$ for environment inference. Ground-truth environment indicators are provided for IRM, OOD-TV-IRM-$\ell_2$, IRM-TV-$\ell_1$, and OOD-TV-IRM-$\ell_1$. Categorical variables, excluding race and sex, are encoded using one-hot encoding, followed by principal component analysis (PCA), retaining over 99% of the cumulative explained variance. The transformed features are then combined with the six integer

---

[1]https://archive.ics.uci.edu/dataset/2/adult

variables, resulting in 59-dimensional representations, which are normalized to have zero mean and unit variance for invariant feature learning.

Table 3 (right) shows the accuracies of different methods on this income prediction task. The proposed OOD-TV-based methods outperform their counterparts in all the 8 cases with both mean and worst accuracies. Hence they are more effective and robust than the original IRM-TV methods in this prediction task regardless of whether the environment partition is available or not.

## 4.5 HOUSE PRICE PREDICTION

The above four experiments are all classification tasks. We also perform a regression task with the House Prices data set[2]. This task uses 15 variables, such as the number of bathrooms, locations, and other similar features, to predict the house price. The training and test sets consist of samples with built years in periods $[1900, 1950]$ and $(1950, 2000]$, respectively. The house prices within the same built year are normalized. The built year variable is fed into EIIL, TIVA, ZIN, OOD-TV-Minimax-$\ell_2$, Minimax-TV-$\ell_1$, and OOD-TV-Minimax-$\ell_1$ for environment inference. LfF is not suitable for this regression task due to the unstable training process and poor performance. The training set is partitioned into 5 subsets with 10-year range in each subset. Samples in the same subset can be seen as having the same environment.

Table 4 (left) provides the mean squared errors (MSE) of different methods in this regression task. The OOD-TV-based methods achieves lower MSEs than their counterparts in the mean result of all the environments and the worst environment. Hence they are more effective and robust than the original IRM-TV methods in the regression task.

Table 4: Mean squared errors of different methods on house price prediction (left) and accuracies of different methods on Colored MNIST (right).

| Method | Mean ± STD | Worst ± STD |
|---|---|---|
| EIIL | $0.3644 \pm 0.0512$ | $0.4953 \pm 0.0587$ |
| TIVA | $0.3869 \pm 0.0413$ | $0.5519 \pm 0.0508$ |
| ZIN | $0.3203 \pm 0.0378$ | $0.4465 \pm 0.0552$ |
| OOD-TV-Minimax-$\ell_2$ | $\mathbf{0.2743 \pm 0.0438}$ | $\mathbf{0.3983 \pm 0.0586}$ |
| Minimax-TV-$\ell_1$ | $0.3339 \pm 0.0508$ | $0.4741 \pm 0.0520$ |
| OOD-TV-Minimax-$\ell_1$ | $\mathbf{0.2621 \pm 0.0510}$ | $\mathbf{0.3706 \pm 0.0549}$ |
| IRM | $0.4589 \pm 0.0762$ | $0.6378 \pm 0.0873$ |
| OOD-TV-IRM-$\ell_2$ | $\mathbf{0.3549 \pm 0.0519}$ | $\mathbf{0.4772 \pm 0.0591}$ |
| IRM-TV-$\ell_1$ | $0.4009 \pm 0.0355$ | $0.5642 \pm 0.0615$ |
| OOD-TV-IRM-$\ell_1$ | $\mathbf{0.3383 \pm 0.0303}$ | $\mathbf{0.4763 \pm 0.0428}$ |

| Method | Mean ± STD |
|---|---|
| EIIL | $0.8236 \pm 0.0265$ |
| LfF | $0.8964 \pm 0.0250$ |
| ZIN | $0.9502 \pm 0.0153$ |
| OOD-TV-Minimax-$\ell_2$ | $\mathbf{0.9671 \pm 0.0060}$ |
| Minimax-TV-$\ell_1$ | $0.9322 \pm 0.0362$ |
| OOD-TV-Minimax-$\ell_1$ | $\mathbf{0.9514 \pm 0.0242}$ |

## 4.6 COLORED MNIST

We use the Colored MNIST data set[3] to evaluate the performance of our approach in multi-group classification of a more general scenario. It consists of hand-written digits of 0 to 9 with different background colors of red, green, and blue. To make it more challenging, we remove the background color information of samples, so that only the mean values of the red, green, and blue channels of a sample can be used as three auxiliary variables for environment inference. EIIL, LfF, OOD-TV-Minimax, and Minimax-TV are suitable for this setting, while TIVA is unavailable due to unknown auxiliary feature extractions. Their classification accuracies are shown in Table 4 (right). The two OOD-TV-Minimax methods achieve the highest accuracies among all the compared methods in this multi-group classification task. Thus OOD-TV-Minimax is effective in improving OOD generalization for such a complex scenario.

## 4.7 NICO

The NICO data set (He et al., 2021) is a widely-used benchmark in Non-Independent and Identically Distributed (Non-I.I.D.) image classification with contexts. This data set presents challenges due to both correlation shift and diversity shift. There are two superclasses in NICO: *Vehicle* and *Animal*,

---

[2]https://www.kaggle.com/c/house-prices-advanced-regression-techniques/data
[3]https://www.kaggle.com/datasets/youssifhisham/colored-mnist-dataset

which consist of 9 classes and 10 classes, respectively. Each class has several contexts, while different classes may have different context sets. We randomly split each context into $80\%$ training samples and $20\%$ test samples. Then we remove the context information of samples and unite all the contexts to form a whole class. Therefore, only the mean values of the red, green, and blue channels of a sample can be used as three auxiliary variables for environment inference. EIIL, OOD-TV-Minimax, and Minimax-TV can be used in this setting, while LfF has unstable training process and poor performance, and TIVA is unavailable due to unknown auxiliary feature extractions. As shown in Table 5, the two OOD-TV-Minimax methods outperform other competitors in terms of classification accuracy. Thus OOD-TV-Minimax is effective in improving OOD generalization with both correlation shift and diversity shift.

Table 5: Accuracies of different methods on NICO.

| Method | Vehicle (Mean $\pm$ STD) | Animal (Mean $\pm$ STD) |
|---|---|---|
| EIIL | $0.6902 \pm 0.0357$ | $0.8664 \pm 0.0110$ |
| ZIN | $0.7176 \pm 0.0374$ | $0.8541 \pm 0.0250$ |
| OOD-TV-Minimax-$\ell_2$ | $\mathbf{0.7872 \pm 0.0275}$ | $\mathbf{0.8826 \pm 0.0249}$ |
| Minimax-TV-$\ell_1$ | $0.7104 \pm 0.0236$ | $0.8289 \pm 0.0311$ |
| OOD-TV-Minimax-$\ell_1$ | $\mathbf{0.7289 \pm 0.0206}$ | $\mathbf{0.9112 \pm 0.0154}$ |

## 4.8 ADVERSARIAL LEARNING PROCESS

To visualize the adversarial learning process, we plot the objective values and parameter changes for OOD-TV-IRM and OOD-TV-Minimax on simulation data in Figure A1. Specifically, the overall objectives are the $g(\Psi, \Phi)$ and $h(\rho, \Psi, \Phi)$ defined in (13) and (14), respectively. The penalty terms correspond to the second terms in (13) and (14), respectively. The parameter changes correspond to $\|\Phi^{(k+1)} - \Phi^{(k)}\|_2$ and $\|\Psi^{(k+1)} - \Psi^{(k)}\|_2$, respectively. We follow the annealing strategy of (Lin et al., 2022) in the early epochs, thus the adversarial learning starts from the 2001st epoch. The Adam scheme (Kingma & Ba, 2015) is adopted as the optimizer. Figure A1 indicates that the adversarial learning process becomes stable with 400 epochs. Both of the overall objective and the penalty term converge to form a stable gap, which corresponds to the fidelity term to be optimized. Besides, both $\|\Phi^{(k+1)} - \Phi^{(k)}\|_2$ and $\|\Psi^{(k+1)} - \Psi^{(k)}\|_2$ converge to small values, which indicates that the parameters $\Phi^{(k)}$ and $\Psi^{(k)}$ are rarely updated.

We also demonstrate the adversarial learning process for OOD-TV-Minimax (since OOD-TV-IRM is not applicable) on Colored MNIST in Figure A2. The Adam optimizer is used for the primal parameter $\Phi$, while the dual update (18, 20) is used for $\Psi$ to ensure convergence. Results show that the adversarial learning process effectively converges after 600 epochs for OOD-TV-Minimax-$\ell_1$ and after 300 epochs for OOD-TV-Minimax-$\ell_2$. All these results indicate that the adversarial learning process is feasible.

## 5 CONCLUSION AND DISCUSSION

We extend the invariant risk minimization based on total variation model (IRM-TV) to a Lagrangian multiplier model OOD-TV-IRM, in order to improve out-of-distribution (OOD) generalization. OOD-TV-IRM is essentially a primal-dual optimization model. The primal optimization minimizes the entire invariant risk to extract invariant features, while the dual optimization strengthens the autonomous TV penalty to provide an adversarial interference. The objective of OOD-TV-IRM is to find a semi-Nash equilibrium that balances the training loss and OOD generalization. We further develop a convergent primal-dual algorithm to facilitate adversarial learning for invariant features and adverse environments. Experimental results show that the proposed OOD-TV-IRM framework improves effectiveness and robustness in most experimental tasks, achieving competitive mean accuracies, worst accuracies, and mean squared errors across different test environments.

One potential future direction is to maximize the benefits of OOD-TV-IRM by improving the diversity and representation of training environments. Another potential approach is to improve the effectiveness of the primal-dual optimization scheme. A third potential approach is to develop new penalties with different variations beyond the widely-used TV penalty.

## ACKNOWLEDGMENTS

This work is supported in part by the National Natural Science Foundation of China under grant 62176103, and in part by the Science and Technology Planning Project of Guangzhou under grant 2024A04J9896.

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

# A   APPENDIX

## A.1   PROOF OF THEOREM 2

*Proof.* **Part (a).**

First, we verify that there exists at least one solution to (15). Since $\lambda(\Psi, \Phi)$ is continuous w.r.t. $(\Psi, \Phi)$ and $(\Psi, \Phi)$ belongs to a bounded and closed set, a solution to $\max_\Psi \lambda(\Psi, \Phi)$ can be obtained for any given $\Phi$ by the Weierstrass extreme value theorem, denoted as $\Psi_{max}(\Phi)$. Let $\Lambda(\Phi) := \lambda(\Psi_{max}(\Phi), \Phi)$. We then turn to prove that $\Lambda(\Phi)$ is a continuous function w.r.t. $\Phi$. For $\Phi_1 \neq \Phi_2$, we have

$$\lambda(\Psi_{max}(\Phi_2), \Phi_1) \leqslant \lambda(\Psi_{max}(\Phi_1), \Phi_1). \tag{21}$$

From the continuity of $\lambda$, for any given $\epsilon > 0$, there exists some $\delta_1 > 0$, such that for any $\Phi \in \mathrm{B}(\Phi_1, \delta_1)$, we have

$$\lambda(\Psi_{max}(\Phi_2), \Phi) - \epsilon < \lambda(\Psi_{max}(\Phi_2), \Phi_1), \tag{22}$$

where $\mathrm{B}(\Phi_1, \delta_1)$ denotes an open ball centered at $\Phi_1$ with radius $\delta_1$. Letting $\Phi_2 \in \mathrm{B}(\Phi_1, \delta_1)$ in (22) and combining it with (21), we have

$$\lambda(\Psi_{max}(\Phi_2), \Phi_2) - \epsilon < \lambda(\Psi_{max}(\Phi_1), \Phi_1). \tag{23}$$

Following a similar deduction, for the above $\epsilon > 0$, there exists some $\delta_2 > 0$, such that for $\Phi_1 \in \mathrm{B}(\Phi_2, \delta_2)$, we have

$$\lambda(\Psi_{max}(\Phi_1), \Phi_1) - \epsilon < \lambda(\Psi_{max}(\Phi_2), \Phi_2). \tag{24}$$

Combining (23) and (24), for any given $\epsilon > 0$, as long as $\|\Phi_1 - \Phi_2\|_2 < \min\{\delta_1, \delta_2\}$, we have

$$|\lambda(\Psi_{max}(\Phi_1), \Phi_1) - \lambda(\Psi_{max}(\Phi_2), \Phi_2)| < \epsilon. \tag{25}$$

This proves that $\Lambda(\Phi)$ is continuous w.r.t. $\Phi$.

As for $(\mathbb{E}_w[|\nabla_w R(w \circ \Phi)|])^2$, the TV operator and the expectation are taken on the variable $w$, which does not affect the continuity w.r.t. $\Phi$. Define

$$G(\Phi) := g(\Psi_{max}(\Phi), \Phi) = \mathbb{E}_w[R(w \circ \Phi)] + \Lambda(\Phi)(\mathbb{E}_w[|\nabla_w R(w \circ \Phi)|])^2. \tag{26}$$

It can be seen that $G(\Phi)$ is the sum, multiplication, and composition of continuous functions w.r.t. $\Phi$, thus it is also continuous w.r.t. $\Phi$. Using the Weierstrass extreme value theorem again, there exists a solution to $\min_\Phi G(\Phi)$, denoted by $\Phi^*$. Let $\Psi^* := \Psi_{max}(\Phi^*)$, then $(\Psi^*, \Phi^*)$ is a solution to (15).

**Part (b).**

Next, we turn to investigate $h(\rho, \Psi, \Phi)$ in (16). From Appendix A.3 of IRM-TV (Lai & Wang, 2024),

$$\mathbb{E}_{w \leftarrow \rho}[|\nabla_w R(w \circ \Phi)|] = \sum_{i=1}^E |\nabla_w R(w \circ \Phi, e_i)| \rho_i, \tag{27}$$

where $e_i$ denotes the $i$-th training environment in the literature of IRM, and $\rho_i$ denotes the environment inference operator for the $i$-th training environment. (27) indicates that $\mathbb{E}_{w \leftarrow \rho}[|\nabla_w R(w \circ \Phi)|]$ is a linear function w.r.t. $\rho \in \Delta^E$, where $\Delta^E$ is the $E$-dimensional simplex defined in Section 3.4. Thus $\mathbb{E}_{w \leftarrow \rho}[|\nabla_w R(w \circ \Phi)|]$ as well as $(\mathbb{E}_{w \leftarrow \rho}[|\nabla_w R(w \circ \Phi)|])^2$ is continuous w.r.t. $\rho$. On the other hand, $\lambda(\Psi, \Phi)$ is continuous w.r.t. $\Psi$. Define

$$\mathfrak{E}(\rho, \Phi) := (\mathbb{E}_{w \leftarrow \rho}[|\nabla_w R(w \circ \Phi)|])^2. \tag{28}$$

To examine the continuity of $\lambda(\Psi, \Phi)\mathfrak{E}(\rho, \Phi)$ w.r.t. the entire dual variable $(\rho, \Psi)$, we check its values at two different points $(\rho_1, \Psi_1)$ and $(\rho_2, \Psi_2)$:

$$\begin{aligned}
&|\lambda(\Psi_1, \Phi)\mathfrak{E}(\rho_1, \Phi) - \lambda(\Psi_2, \Phi)\mathfrak{E}(\rho_2, \Phi)| \\
<&|\lambda(\Psi_1, \Phi)\mathfrak{E}(\rho_1, \Phi) - \lambda(\Psi_1, \Phi)\mathfrak{E}(\rho_2, \Phi)| + |\lambda(\Psi_1, \Phi)\mathfrak{E}(\rho_2, \Phi) - \lambda(\Psi_2, \Phi)\mathfrak{E}(\rho_2, \Phi)| \\
=&|\lambda(\Psi_1, \Phi)| \cdot |\mathfrak{E}(\rho_1, \Phi) - \mathfrak{E}(\rho_2, \Phi)| + |\mathfrak{E}(\rho_2, \Phi)| \cdot |\lambda(\Psi_1, \Phi) - \lambda(\Psi_2, \Phi)|.
\end{aligned} \tag{29}$$

When $(\rho_2, \Psi_2) \to (\rho_1, \Psi_1)$, the right hand side of (29) tends to zero, thus the left hand side of (29) also tends to zero. This verifies that $\lambda(\Psi, \Phi)\mathfrak{E}(\rho, \Phi)$ is continuous w.r.t. $(\rho, \Psi)$ for any given $\Phi$.

Using the Weierstrass extreme value theorem, there exists at least one solution to $\max_{\rho, \Psi}[\lambda(\Psi, \Phi)\mathfrak{E}(\rho, \Phi)]$ for any given $\Phi$, denoted by $(\rho, \Psi)_{max}(\Phi)$. Define

$$H(\Phi) := h((\rho, \Psi)_{max}(\Phi), \Phi) = \mathbb{E}_{w \leftarrow \frac{1_{(E)}}{E}}[R(w \circ \Phi)] + \max_{\rho, \Psi}[\lambda(\Psi, \Phi)\mathfrak{E}(\rho, \Phi)], \qquad (30)$$

which is continuous w.r.t. $\Phi$. Following similar deductions to Part (a), there exists a solution to $\min_\Phi H(\Phi)$, denoted by $\Phi^*$. Let $(\rho^*, \Psi^*) := (\rho, \Psi)_{max}(\Phi^*)$, then $(\rho^*, \Psi^*, \Phi^*)$ is a solution to (16).

**Part (c).**

Lastly, we turn to verify that any solutions $(\Psi^*, \Phi^*)$ to (15) and $(\rho^*, \Psi^*, \Phi^*)$ to (16) are semi-Nash equilibria. We first fix $\Phi^*$, since $\Psi^* = \Psi_{max}(\Phi^*)$ and $(\rho^*, \Psi^*) = (\rho, \Psi)_{max}(\Phi^*)$, we have $g(\Psi^*, \Phi^*) \geqslant g(\Psi, \Phi^*)$ or $h(\rho^*, \Psi^*, \Phi^*) \geqslant h(\rho, \Psi, \Phi^*)$ for any $\Psi$ or $(\rho, \Psi)$ in the parameter space. Hence Item 1 in Definition 1 is satisfied.

On the other hand, from (26) and (30), we have $G(\Phi^*) \leqslant G(\Phi)$ and $H(\Phi^*) \leqslant H(\Phi)$ in the parameter space. Hence Item 2 in Definition 1 is also satisfied. In summary, Theorem 2 is proved.

$\square$

## A.2 PROOF OF THEOREM 3

Before starting the proof, we compute some gradients or subgradients in place of the conventional gradients at non-differentiable points for $g$ and $h$. This allows us to maintain the optimization flow with the autograd module of mainstream learning architectures (like Pytorch[4]). First, the subgradient of $|\nabla_w R(w \circ \Phi)|$ w.r.t. $\Phi$ is:

$$\partial_\Phi |\nabla_w R(w \circ \Phi)| = \begin{cases} \frac{J_\Phi^\top[\nabla_w R(w \circ \Phi)] * \nabla_w R(w \circ \Phi)}{|\nabla_w R(w \circ \Phi)|} & \text{if} \quad \nabla_w R(w \circ \Phi) \neq 0, \\ 0 & \text{if} \quad \nabla_w R(w \circ \Phi) = 0, \end{cases} \qquad (31)$$

where $J_\Phi[\cdot]$ is the Jacobian matrix w.r.t. $\Phi$, and $*$ is the matrix multiplication.

By the chain rule of derivative, the (sub)gradients of $g$ w.r.t. $\Phi$ and $\Psi$ are:

$$\begin{aligned} \partial_\Phi g(\Psi, \Phi) =& \mathbb{E}_w[\nabla_\Phi R(w \circ \Phi)] + 2\lambda(\Psi, \Phi) \cdot \mathbb{E}_w[|\nabla_w R(w \circ \Phi)|] \cdot \mathbb{E}_w[\partial_\Phi |\nabla_w R(w \circ \Phi)|] \\ & + \nabla_\Phi \lambda(\Psi, \Phi) \cdot (\mathbb{E}_w[|\nabla_w R(w \circ \Phi)|])^2, \\ \nabla_\Psi g(\Psi, \Phi) =& \nabla_\Psi \lambda(\Psi, \Phi) \cdot (\mathbb{E}_w[|\nabla_w R(w \circ \Phi)|])^2. \end{aligned} \qquad (32)$$

Similarly, the (sub)gradients of $h$ w.r.t. $\rho$, $\Psi$, and $\Phi$ are:

$$\begin{aligned} \nabla_\rho h(\rho, \Psi, \Phi) =& 2\lambda(\Psi, \Phi) \cdot \mathbb{E}_{w \leftarrow \rho}[|\nabla_w R(w \circ \Phi)|] \cdot \nabla_\rho(\mathbb{E}_{w \leftarrow \rho}[|\nabla_w R(w \circ \Phi)|]), \\ \nabla_\Psi h(\rho, \Psi, \Phi) =& \nabla_\Psi \lambda(\Psi, \Phi) \cdot (\mathbb{E}_{w \leftarrow \rho}[|\nabla_w R(w \circ \Phi)|])^2, \\ \partial_\Phi h(\rho, \Psi, \Phi) =& \mathbb{E}_{w \leftarrow \frac{1_{(E)}}{E}}[\nabla_\Phi R(w \circ \Phi)] \\ & + 2\lambda(\Psi, \Phi) \cdot \mathbb{E}_{w \leftarrow \rho}[|\nabla_w R(w \circ \Phi)|] \cdot \mathbb{E}_{w \leftarrow \rho}[\partial_\Phi |\nabla_w R(w \circ \Phi)|] \\ & + \nabla_\Phi \lambda(\Psi, \Phi) \cdot (\mathbb{E}_{w \leftarrow \rho}[|\nabla_w R(w \circ \Phi)|])^2. \end{aligned} \qquad (33)$$

*Proof of Theorem 3.* We first investigate the sequence $\{\Phi^{(k)}\}$. Combining (17) and (19), we have

$$\|\Phi^{(k+1)} - \Phi^{(k)}\|_2 \leqslant \frac{1}{k^p}. \qquad (34)$$

Summing up both sides of (34) from $k = 2$ to $\infty$ yields

$$\sum_{k=2}^\infty \|\Phi^{(k+1)} - \Phi^{(k)}\|_2 \leqslant \sum_{k=2}^\infty \frac{1}{k^p} < \int_1^\infty \frac{1}{\zeta^p} \, d\zeta = \frac{1}{1-p} \cdot \frac{1}{\zeta^{p-1}}\Big|_1^\infty = \frac{1}{p-1} < \infty. \qquad (35)$$

---

[4] https://pytorch.org/

It means that the sum of infinite series $\sum_{k=2}^{\infty} \|\Phi^{(k+1)} - \Phi^{(k)}\|_2$ is convergent. Then for any given $\epsilon > 0$, there exists some $N$ such that $\sum_{k=N}^{\infty} \|\Phi^{(k+1)} - \Phi^{(k)}\|_2 < \epsilon$. Hence for any $k > N$ and any $m \in \mathbb{N}_+$,

$$\|\Phi^{(k+m)} - \Phi^{(k)}\|_2 \leqslant \sum_{j=k}^{k+m-1} \|\Phi^{(j+1)} - \Phi^{(j)}\|_2 \leqslant \sum_{j=N}^{\infty} \|\Phi^{(j+1)} - \Phi^{(j)}\|_2 < \epsilon. \tag{36}$$

It indicates that $\{\Phi^{(k)}\}$ is a Cauchy sequence. From the completeness of the parameter space for $\Phi$, $\Phi^{(k)}$ converges to some point $\Phi^{\bullet}$. Following similar deductions, the sequences $\{\Psi^{(k)}\}$ associated with $g$, $\{\Phi^{(k)}\}$ associated with $h$, and $\{(\rho^{(k)}, \Psi^{(k)})\}$ associated with $h$ also converge.

Next, we turn to analyze the computational complexity of this primal-dual algorithm. Without loss of generality, we consider one iteration as having a complexity of $\mathcal{O}(1)$, since the complexity within one iteration depends on specific forms of $g$ or $h$ and is relatively constant across different (sub)gradient-type algorithms. Suppose the algorithm needs $N$ iterations in order to achieve a convergence tolerance of $\epsilon$. From the above deduction, we have

$$\|\Phi^{(N)} - \Phi^{\bullet}\|_2 \leqslant \sum_{k=N}^{\infty} \|\Phi^{(k+1)} - \Phi^{(k)}\|_2 < \int_{N-1}^{\infty} \frac{1}{\zeta^p} \, \mathrm{d}\zeta = \frac{1}{p-1} \cdot \frac{1}{(N-1)^{p-1}} < \epsilon. \tag{37}$$

The last inequality in (37) indicates that the convergence tolerance $\epsilon$ can be achieved as long as

$$N > ((p-1)\epsilon)^{-\frac{1}{p-1}} + 1. \tag{38}$$

By taking the smallest integer for $N$ in (38), we obtain the computational complexity $\mathcal{O}(((p-1)\epsilon)^{-\frac{1}{p-1}})$.

$\square$

### A.3 SIMULATION DATA GENERATION

The binary outcome of interest $Y(t)$ with time $t \in [0, 1]$ is caused and influenced by the invariant features $X_v(t) \in \mathbb{R}$ and the spurious features $X_s(t) \in \mathbb{R}$:

$$X_v(t) \sim \begin{cases} \mathcal{N}(1, 1), \text{ w.p. } 0.5; \\ \mathcal{N}(-1, 1), \text{ w.p. } 0.5. \end{cases} \qquad Y(t) \sim \begin{cases} \text{sign}(X_v(t)), & \text{w.p. } p_v; \\ -\text{sign}(X_v(t)), & \text{w.p. } 1 - p_v. \end{cases}$$

$$X_s(t) \sim \begin{cases} \mathcal{N}(Y(t), 1), & \text{w.p. } p_s(t); \\ \mathcal{N}(-Y(t), 1), & \text{w.p. } 1 - p_s(t). \end{cases}$$

$X_v(t)$ and $X_s(t)$ are further extended to 5 and 10 dimensional sequences by adding standard Gaussian noise, respectively. We control the correlation between $X_v(t)$ and $Y(t)$ by a parameter $p_v$, keeping it constant, but change the correlation between $X_s(t)$ and $Y(t)$ as a function of $t$ over the interval $[0, 1]$ by a parameter $p_s(t)$. Hence, the training data is generated with parameter settings $(p_s^-, p_s^+, p_v)$, where $p_s^-$ and $p_s^+$ denote the $p_s(t)$ setting for $t \in [0, 0.5)$ and $t \in [0.5, 1]$, respectively. As for the test data, we set $p_s \in \{0.999, 0.8, 0.2, 0.001\}$ and keep the same $p_v$. Time $t$ can be exploited as the auxiliary variable in EIIL, LfF, TIVA, ZIN, OOD-TV-Minimax-$\ell_2$, Minimax-TV-$\ell_1$, and OOD-TV-Minimax-$\ell_1$ for environment inference.

### A.4 MORE RESULTS OF ADVERSARIAL LEARNING PROCESS

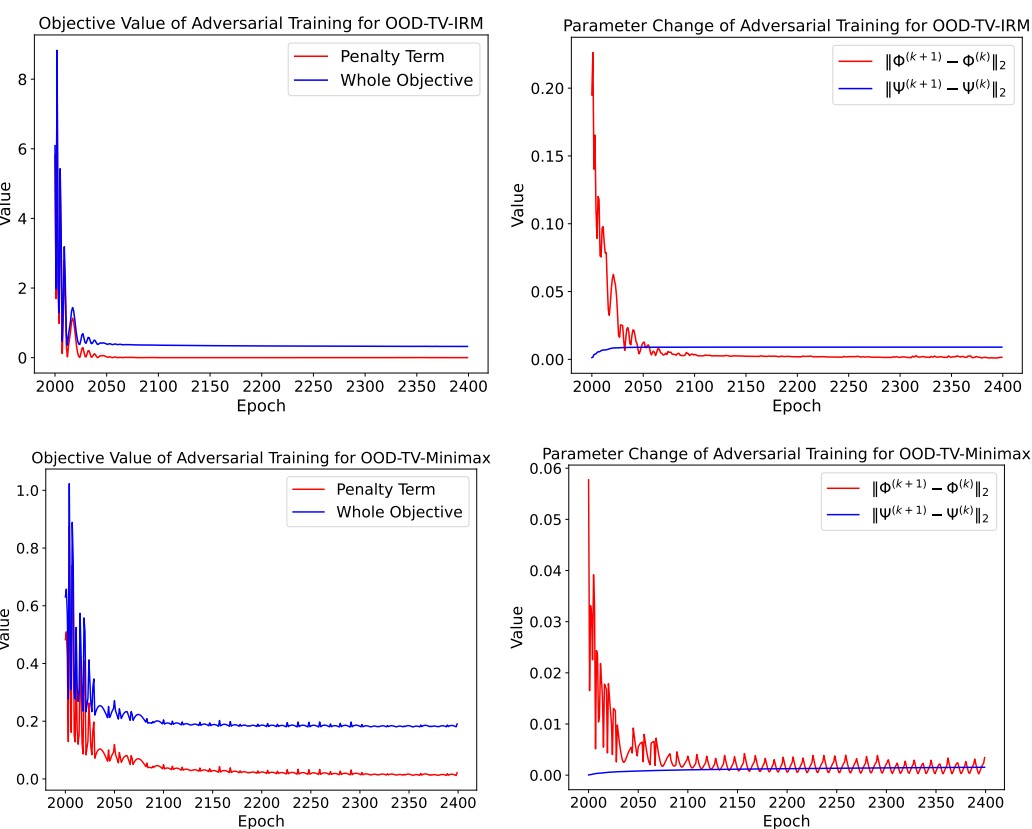

Figure A1: Objective value and parameter change of adversarial learning for OOD-TV-IRM and OOD-TV-Minimax on simulation data.

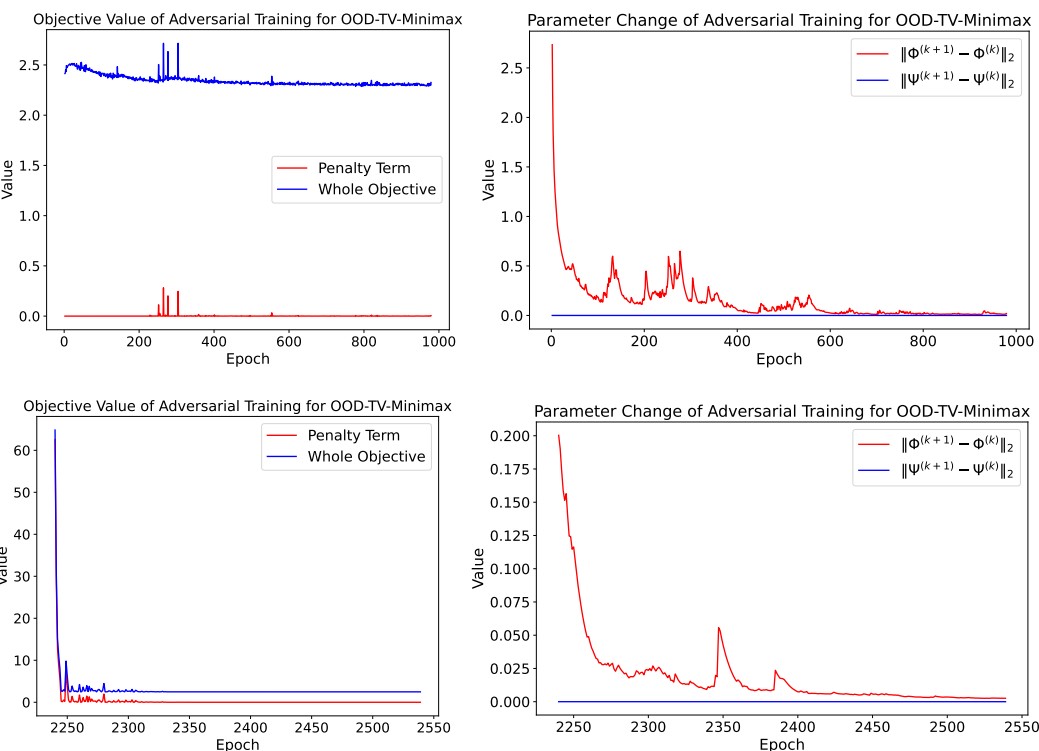

Figure A2: Objective value and parameter change of adversarial learning for OOD-TV-Minimax on the Colored MNIST data set. The top two figures correspond to OOD-TV-Minimax-$\ell_1$, while the bottom two figures correspond to OOD-TV-Minimax-$\ell_2$.

