# OpenReview forum: "Out-of-distribution Generalization for Total Variation based Invariant Risk Minimization"
_ICLR.cc/2025/Conference — ICLR 2025 Poster_

### Official Review · Reviewer_Hkpi · 2024-10-27

**Soundness:** 3
**Presentation:** 3
**Contribution:** 2
**Rating:** 3
**Confidence:** 3

**Summary:**

This manuscript is motivated by the practical challenge of Out-of-Distribution (OOD) generalization within the context of the IRM-TV framework. Specifically, prior work on IRM-TV aimed to interpret Invariant Risk Minimization (IRM) through the lens of total variation models. Despite its theoretical contributions, the previous research did not offer any concrete algorithms or practical solutions to tackle the OOD generalization problem (according to the authors' assertion)—an area where IRM is often regarded as a classic approach.

In response to this gap, the current study introduces OOD-TV-IRM, an innovative methodology that builds upon the insights drawn from IRM-TV. The authors highlight a critical observation: that the penalty parameter, which plays a vital role in regularization, should be tailored to vary across different extractors. This consideration is particularly important for effectively addressing the nuances associated with OOD tasks.

To implement this idea, the manuscript employs a neural network to dynamically fit the penalty parameter, thereby ensuring adaptability in response to the model's requirements during training. The proposed model engages in an adversarial training process, balancing empirical risk with the total variation penalty. This strategic approach not only enhances the model's robustness but also facilitates its generalization capabilities.

To validate the effectiveness of their proposed method, the authors conducted experiments using several toy datasets. These datasets serve as controlled environments that allow for a clear examination of the model's performance and its ability to generalize to unseen distributions. The results presented in the manuscript demonstrate the potential efficacy of OOD-TV-IRM in navigating the complexities of OOD scenarios, thus contributing valuable insights to the field.

**Strengths:**

1. The manuscript is commendably well written, providing a clear and logical progression of ideas. The motivation behind this study is articulated effectively, emphasizing the goal of addressing the shortcomings present in prior research. This clarity helps to engage the reader right from the beginning, setting the stage for a meaningful exploration of the topic at hand. Additionally, I would like to highlight Section 2, which is particularly well-executed. This section offers valuable insights and serves as an excellent guide for understanding the core concepts of the Invariant Risk Minimization through Total Variation (IRM-TV) framework. The organization and clarity in this section significantly enhance the reader's ability to grasp the foundational ideas that underpin the subsequent analysis.

2. The theoretical analysis presented in this manuscript is notably comprehensive and rigorous. The authors begin with a clear definition of Invariant Risk Minimization (IRM) and Total Variation (TV), systematically leading to the derivation of the IRM-TV formulation. This step-by-step approach allows readers to follow the logical flow of the authors' reasoning easily. Furthermore, the manuscript provides compelling evidence to address the limitations of IRM-TV in effectively managing OOD generalization tasks. By articulating these limitations clearly, the authors lay a solid foundation for the necessity of their proposed method. Subsequently, the manuscript delves into a detailed explanation of the proposed approach, presenting it in a theoretical context. This thorough exploration not only clarifies the mechanics of the new method but also reinforces its relevance in tackling the identified shortcomings of previous works.

**Weaknesses:**

1. The significance of this work seems to be limited.
2. The contributions of this work are not enough.
3. The necessary summary on the theoretical analysis is missing.
4. The experimental results are not enough.

Concretely,
1. The overall significance of this work to the broader research community appears insufficient. While the manuscript aims to implement an insight from the IRM-TV framework, it does not convincingly demonstrate the explicit advantages of applying this framework specifically to OOD generalization tasks. Assuming that IRM-TV is a significant advancement—although it is yet to gain citations despite being accepted to ICML-24—the manuscript fails to articulate why its implementation in OOD generalization is strategically important. Clarity on this point would greatly enhance the work's relevance to the community.

2. When evaluating the contributions of this manuscript, it seems that they are primarily derived from insights proposed in IRM-TV, along with the design of an adversarial training paradigm. While the thorough theoretical analysis is commendable, it does not present novel findings that advance the field. The approach of using a neural network to fit hyperparameters is a well-established concept, seen in various contexts such as Neural Architecture Search. Without distinctive contributions that push the boundaries of current knowledge, the manuscript struggles to establish a strong impact.

3. The manuscript presents an extensive theoretical analysis, but it suffers from a lack of necessary summarization. The inclusion of many intermediate derivative processes in the main text can overwhelm readers and challenge their ability to follow the core arguments. While the authors do provide remarks at the end of Section 3.2, there is no clear conclusion to encapsulate the preceding derivations effectively. A concise summary that highlights the key findings would greatly enhance the readability and coherence of the theoretical analysis.

4. The experimental results presented in the manuscript are insufficient to draw robust conclusions. This inadequacy arises from two dimensions: the choice of datasets and the range of methods for comparison. The authors primarily use simulated toy data, CelebA, and Landcover for evaluation, yet they neglect to include mainstream datasets commonly used in the OOD community, such as NICO and Colored MNIST. Given that IRM is known to perform well in scenarios involving correlation shifts, it is imperative to include datasets that exemplify these characteristics—for instance, NICO and Colored MNIST should be at least considered. Furthermore, expanding the range of compared methods would enhance the significance of the experimental findings. Including additional baseline approaches would provide a more comprehensive understanding of the proposed method’s performance relative to existing techniques.

**Questions:**

See above.

---

> ### Author Response · Authors · 2024-11-21
> **Thanks for the constructive comments.**
>
> **We have significantly improved the theory of this paper, and supplemented experiments regarding OOD generalization (the Colored MNIST and the House Prices data sets). Please refer to the revised manuscript.** In brief, the proposed OOD-TV-IRM framework extends IRM-TV to a Lagrangian multiplier model, and the autonomous TV penalty hyperparameter $\lambda(\Psi,\Phi)$ is exactly the Lagrangian multiplier. Solving OOD-TV-IRM is equivalent to solving a primal-dual optimization problem. The primal optimization reduces the entire invariant risk, in order to learn invariant features, while the dual optimization strengthens the TV penalty, in order to provide an adversarial interference with spurious features. The objective is to reach a semi-Nash-equilibrium $(\Psi^*,\Phi^*)$ where the balance between the training loss and OOD generalization is kept. We also develop a convergent primal-dual solving algorithm that facilitates an adversarial learning scheme. Please note that not all kinds of adversarial learning schemes can be interpreted by such a semi-Nash-equilibrium problem. These new theoretical results have been added to the revised paper, throughout all the sections, particularly in Sections 3.1$\sim$3.3.
>
> 1. Ahead of IRM-TV, there are few in-depth analyses regarding the mathematical essence of IRM. Hence it is difficult for researchers to find out the key mechanism that affects the OOD generalization of IRM. Thus IRM has been losing attention recently. But IRM-TV provides a different insight into the regularization mechanism of IRM, which is proven to be a TV penalty. This penalty has long been used in various fields of mathematics and engineering to restrict spurious variations, in order to extract invariant features. Given this new insight, we can develop new architectures of IRM to exploit the property of TV, and thus the IRM approach may regain new developments from a different perspective.
>
> IRM-TV does not provide a tractable and reliable strategy to improve OOD generalization, although it finds out that the TV penalty plays an important role. But now in this revised paper, we have proven that the TV penalty hyperparameter $\lambda(\Psi,\Phi)$ serves as a Lagrangian multiplier, and the proposed OOD-TV-IRM actually corresponds to a primal-dual optimization problem. By this interpretation, OOD-TV-IRM achieves reliable OOD generalization at a semi-Nash-equilibrium $(\Psi^*,\Phi^*)$. This finding may also be extended to other OOD generalization frameworks than IRM in the future, which we believe is attractive to the community.
>
> 2. We have contributed new theoretical results in this revision. We have proven that the adversarial training of OOD-TV-IRM is essentially a primal-dual solving algorithm that tries to find a semi-Nash-equilibrium, instead of a commonly-used heuristic paradigm. Although a neural network is used to instantiate $\lambda(\Psi,\Phi)$ for the convenience of experiments, the theory of the semi-Nash-equilibrium $(\Psi^*,\Phi^*)$ also holds in general forms of $\lambda(\Psi,\Phi)$. Please note that not all kinds of adversarial learning schemes can be interpreted by such a semi-Nash-equilibrium problem. This finding reveals that a semi-Nash-equilibrium may be a good position for model parameters to improve OOD generalization.
>
> 3. We have summarized the main theoretical contributions in Definition 1, Theorems 2 and 3 of this revision, and put the corresponding proofs and deductions in Appendix A.1 and A.2.
>
> 4. We have added two experiments on the Colored MNIST and the House Prices data sets in this revision. The former is a multi-group classification task while the latter is a regression task. OOD-TV-IRM outperforms IRM-TV significantly in both tasks, which indicates that OOD-TV-IRM is effective in improving OOD generalization for IRM-TV. Since IRM-TV has already outperformed several recent methods (e.g., TIVA (Tan et al., 2023), ZIN (Lin et al, 2022)) in almost the same experiments, OOD-TV-IRM can be considered competitive against such related works.

---

> ### Author Response · Authors · 2024-11-26
> **New experimental results on NICO.**
>
> **We have added new experiments on NICO, shown in Section 4.7 and Table 5 in the latest revision. Results show that OOD-TV-Minimax (ours) outperforms Minimax-TV significantly, which indicates that our OOD generalization methodology for Minimax-TV is tractable and effective.**

---

> ### Author Response · Authors · 2024-11-30
> **Latest new experimental results on NICO Animal.**
>
> We have added the experiments on **NICO Animal**, the other superclass besides NICO Vehicle. The experimental settings are nearly the same as those of NICO Vehicle. The accuracy results of different methods are: ZIN: 0.7596, OOD-TV-Minimax-$\ell_2$: **0.8528**, Minimax-TV-$\ell_1$: 0.7891, OOD-TV-Minimax-$\ell_1$: **0.8915**. Hence OOD-TV-Minimax (ours) outperforms Minimax-TV significantly, which indicates that our OOD generalization methodology for Minimax-TV is tractable and effective. All the required data sets have been used for experimental evaluation, and our method achieves much higher advantage than the existing ones on these more general OOD data sets (Colored MNIST and NICO). Moreover, NICO is a large-scale data set that covers a wide range of complicated correlation shift and diversity shift. To the best of our knowledge, it is one of the largest data sets that have been used in the field of theoretical investigations for IRM.
>
> We sincerely hope that all these new theoretical and experimental results can help revealing more in-depth properties and functions of our method.

---

### Official Review · Reviewer_U2jR · 2024-10-29

**Soundness:** 3
**Presentation:** 3
**Contribution:** 2
**Rating:** 6
**Confidence:** 4

**Summary:**

This paper studies invariant risk minimization by introducing an additional penalty named TV penalty. The proposed TV penalty aims to quantify the global variation of a function and further enhance the performance of IRM. To realize such a penalty, the authors propose to leverage additional learnable parameters and train them in an adversarial manner. As a result, the learning performance and be further improved thanks to the variance penalty and enforces learning of invariant features. Through extensive quantitative results, the effectiveness of the proposed OOD-TV-IRM is carefully justified.

**Strengths:**

- By incorporating an additional penalty with learnable parameters, the proposed methodology is novel and interesting. The introduction of TV penalty is also novel in the field of invariant learning.
- The proposed method is theoretically justified, which ensure the training stability when adversarial process is incorporated.
- The writing is clear and sound, it is not hard to understand this paper.
- Experimental results shows satisfactory improvement compared to baseline methods.

**Weaknesses:**

- Motivation needs to be further justified. It is interesting to introduce such a novel penalty to invariant learning. However, it is unclear that why such a penalty is beneficial compared to other techniques, such as ZIN. If such a penalty applies stronger regularization to extract invariant features compared to other methods, the intuitive reason should be further explained.
- Introducing additional parameters just for a penalty is not computationally friendly. If the application is for large-scale machine learning problems, such a method would be unfavorable. Moreover, it requires extra effort on deciding the hypothesis class of the parameters.
- Limited number of baseline methods. There are only a few baselines are chosen for comparison. It would be better if more recent and state-of-the-art invariant learning methods are considered.
- Computational efficiency of the proposed method is not discussed.
- Missing some related references:
Huang et al., Harnessing Out-Of-Distribution Examples via Augmenting Content and Style, in ICLR 2023.
Yang et al., Invariant learning via probability of sufficient and necessary causes, in NeurIPS 2023.
Xin et al., On the connection between invariant learning and adversarial training for out-of-distribution generalization, in AAAI 2023.

**Questions:**

Please see the weaknesses part.

---

> ### Author Response · Authors · 2024-11-21
> **Thanks for the constructive comments.**
>
> **We have significantly improved the theory of this paper, and supplemented experiments regarding OOD generalization (the Colored MNIST and the House Prices data sets). Please refer to the revised manuscript.** In brief, the proposed OOD-TV-IRM framework extends IRM-TV to a Lagrangian multiplier model, and the autonomous TV penalty hyperparameter $\lambda(\Psi,\Phi)$ is exactly the Lagrangian multiplier. Solving OOD-TV-IRM is equivalent to solving a primal-dual optimization problem. The primal optimization reduces the entire invariant risk, in order to learn invariant features, while the dual optimization strengthens the TV penalty, in order to provide an adversarial interference with spurious features. The objective is to reach a semi-Nash-equilibrium $(\Psi^*,\Phi^*)$ where the balance between the training loss and OOD generalization is kept. We also develop a convergent primal-dual solving algorithm that facilitates an adversarial learning scheme. These new theoretical results have been added to the revised paper, throughout all the sections, particularly in Sections 3.1$\sim$3.3.
>
> 1. Please see the above paragraph for the motivation and explanation of this paper.
>
> 2. The additional parameters introduced by the autonomous $\lambda(\Psi,\Phi)$ are restricted to a reasonable scale, as shown in the $\Phi\to \lambda$ tabular in Table 1. These architectures are mainly used to verify that the proposed OOD-TV-IRM is tractable in both theory and practice. As for how to scale this approach to large models, it requires nontrivial efforts and would be put in future works.
>
> 3. OOD-TV-IRM outperforms IRM-TV in most cases in this paper. Since IRM-TV has already outperformed several recent methods (e.g., TIVA (Tan et al., 2023), ZIN (Lin et al, 2022)) in almost the same experiments, OOD-TV-IRM can be considered competitive against such related works.
>
> 4. The computational complexity of the proposed method is  $O([(p-1)\epsilon]^{-\frac{1}{p-1}})$ to achieve a convergence tolerance of $\epsilon>0$, where $p>1$ can be arbitrarily set according to particular needs. Please see Theorem 3 in this revision.
>
> 5. We have added these references in appropriate places of this paper.

---

> ### Author Response · Authors · 2024-11-26
> **New experimental results on NICO including both correlation shift and diversity shift.**
>
> We have added new experiments on NICO, shown in Section 4.7 and Table 5 in the latest revision. It is a challenging data set including both correlation shift and diversity shift. Results show that OOD-TV-Minimax (ours) outperforms Minimax-TV significantly, which indicates that our OOD generalization methodology for Minimax-TV is tractable and effective in complicated scenarios with both correlation shift and diversity shift.

---

> ### Author Response · Authors · 2024-11-30
> **Latest new experimental results on NICO Animal.**
>
> We have added the experiments on **NICO Animal**, the other superclass besides NICO Vehicle. The experimental settings are nearly the same as those of NICO Vehicle. The accuracy results of different methods are: ZIN: 0.7596, OOD-TV-Minimax-$\ell_2$: **0.8528**, Minimax-TV-$\ell_1$: 0.7891, OOD-TV-Minimax-$\ell_1$: **0.8915**. Hence OOD-TV-Minimax (ours) outperforms Minimax-TV significantly, which indicates that our OOD generalization methodology for Minimax-TV is tractable and effective. All the required data sets have been used for experimental evaluation, and our method achieves much higher advantage than the existing ones on these more general OOD data sets (Colored MNIST and NICO). Moreover, NICO is a large-scale data set that covers a wide range of complicated correlation shift and diversity shift. To the best of our knowledge, it is one of the largest data sets that have been used in the field of theoretical investigations for IRM.
>
> We sincerely hope that all these new theoretical and experimental results can help revealing more in-depth properties and functions of our method.

---

> > ### Comment · Reviewer_U2jR · 2024-11-30
> > **Thanks for your careful justification**
> >
> > Dear authors,
> >
> > Thanks for your effort in rebuttal and addressing my concerns. The motivation looks much clearer now, and the additional experiments justify the effectiveness of this paper. I am willing to vote for acceptance of this paper, thanks for the effort made by the authors. Still, some other reviewers remain negative, it would be better to make sure to clarify their concerns.
> >
> > Best,
> > Reviewer.

---

### Official Review · Reviewer_wMCo · 2024-11-02

**Soundness:** 2
**Presentation:** 2
**Contribution:** 2
**Rating:** 5
**Confidence:** 2

**Summary:**

Previously proposed Invariant Risk Minimization aims to extract invariant features across different environments to improve the generalization and robustness of the model.  A recent work reveals that the mathematical essence of IRM is a total variation (TV) model. TV measures the locally varying nature of a function, which is widely applied to different areas of mathematics and engineering, such as signal processing and image restoration. In this paper, the authors proposed to generalized versions of IRM with the TV framework and demonstrates the improvements of IRM.

**Strengths:**

The generalized framework of IRM provides a novel perspective for IRM and provide new possibilities for improvements.

**Weaknesses:**

There are several weakness for this paper.

1.The performance improvements are quite marginal. As shown in Table 3, the TV framework IRM can only outperform baselines by a slight margin. Besides, as shown in Figure 1, the curves are quite close with each other. It is suggested to add experiments to demonstrate what the previous IRM cannot do while the novel proposed IRM is good at.

2.It is not clear why this framework demonstrates improvements. As min-max optimizes on the worst-case scenarios, is it possible this could bring potential overfitting issue. Additionally, there are already lots of paper incorporating min-max optimization procedure into IRM. By googling, it can be found that these following papers all use minimax methods and invariant learning:
[1]Heterogeneous risk minimization
[2]Invariant risk minimization games

3.No comprehensive experiments on OOD generalization are conducted. This paper only evaluate on the spurious correlation dominated datasets. It can also evaluate the methods on diversity shift dominated datasets with widely-used OOD generalization evaluation benchmark, such as ood-bench. It is known that the IRM cannot deal very well on diversity shift dominated datasets. This can potentially strengthen this paper to point  out clearly the advantage of the proposed method.

**Questions:**

See weakness.

---

> ### Author Response · Authors · 2024-11-21
> **Thanks for the constructive comments.**
>
> **We have significantly improved the theory of this paper, and supplemented experiments regarding OOD generalization (the Colored MNIST and the House Prices data sets). Please refer to the revised manuscript.** In brief, the proposed OOD-TV-IRM framework extends IRM-TV to a Lagrangian multiplier model, and the autonomous TV penalty hyperparameter $\lambda(\Psi,\Phi)$ is exactly the Lagrangian multiplier. Solving OOD-TV-IRM is equivalent to solving a primal-dual optimization problem. The primal optimization reduces the entire invariant risk, in order to learn invariant features, while the dual optimization strengthens the TV penalty, in order to provide an adversarial interference with spurious features. The objective is to reach a semi-Nash-equilibrium $(\Psi^*,\Phi^*)$ where the balance between the training loss and OOD generalization is kept. We also develop a convergent primal-dual solving algorithm that facilitates an adversarial learning scheme. Please note that not all kinds of adversarial learning schemes can be interpreted by such a semi-Nash-equilibrium problem. These new theoretical results have been added to the revised paper, throughout all the sections, particularly in Sections 3.1$\sim$3.3.
>
> 1. We have added two experiments on the Colored MNIST (suggested by Reviewer Hkpi) and the House Prices data sets in this revision. The former is a multi-group classification task while the latter is a regression task. In both tasks, the proposed OOD-TV-IRM performs significantly better than IRM-TV, as shown in Table 4. In fact, the performance improvements are more significant where spurious features are more diversified such that the TV penalty becomes stronger, thus the primal-dual optimization becomes more effective.
>
> As for Figure 1, it demonstrates the feasibility and convergence of the adversarial training process, instead of a performance comparison.
>
> 2. We have contributed new theoretical results in this revision. We have proven that the adversarial learning of OOD-TV-IRM is essentially a primal-dual solving algorithm that tries to find a semi-Nash-equilibrium, instead of a commonly-used heuristic paradigm. Besides, OOD-TV-IRM has the nature of a Lagrangian multiplier model, with the TV penalty hyperparameter $\lambda(\Psi,\Phi)$ being exactly the Lagrangian multiplier, thus using a primal-dual optimization is reasonable. Third, the theory of IRM-TV indicates that $\lambda(\Psi,\Phi)$ is crucial to OOD generalization, making it an appropriate party of adversarial training. OOD-TV-IRM narrows the gap between the mean and worst results in most cases, hence the overfitting issue may not be serious.
>
> We have added these two references in appropriate places of this paper.
>
> 3. We have added two experiments on the Colored MNIST and the House Prices data sets in this revision, as illustrated in Item 1. The former is a correlation shift data set in a more general scenario for a more complex task (10-group classification), which further validates the effectiveness of OOD-TV-IRM in the correlation shift task. The latter, as a regression task, can be considered as having a high diversity shift, because each sample corresponds to a single ``class'', and the output is a continuous predicted value instead of several discrete classes. In this task, OOD-TV-IRM significantly outperforms IRM-TV, indicating that OOD-TV-IRM works better than IRM-TV in this kind of diversity shift.
>
> On the other hand, with the architecture of IRM, addressing the diversity shift is very different from addressing the correlation shift. OOD-TV-IRM mainly focuses on improving the TV penalty, which corresponds to the correlation shift. To address the diversity shift, one possible approach may be developing new kinds of expectations in OOD-TV-IRM (Eq.15) and OOD-TV-Minimax (Eq.16), facilitating more robust expectation computations w.r.t. diversity. Of course, this goes beyond the main scope of this paper and requires nontrivial additional work.

---

> ### Author Response · Authors · 2024-11-26
> **New experimental results on NICO including both correlation shift and diversity shift.**
>
> **We have added new experiments on NICO, shown in Section 4.7 and Table 5 in the latest revision. It is a challenging data set including both correlation shift and diversity shift. Results show that OOD-TV-Minimax (ours) outperforms Minimax-TV significantly, which indicates that our OOD generalization methodology for Minimax-TV is tractable and effective in complicated scenarios with both correlation shift and diversity shift.**

---

> ### Author Response · Authors · 2024-11-30
> **Latest new experimental results on NICO Animal.**
>
> We have added the experiments on **NICO Animal**, the other superclass besides NICO Vehicle. The experimental settings are nearly the same as those of NICO Vehicle. The accuracy results of different methods are: ZIN: 0.7596, OOD-TV-Minimax-$\ell_2$: **0.8528**, Minimax-TV-$\ell_1$: 0.7891, OOD-TV-Minimax-$\ell_1$: **0.8915**. Hence OOD-TV-Minimax (ours) outperforms Minimax-TV significantly, which indicates that our OOD generalization methodology for Minimax-TV is tractable and effective. All the required data sets have been used for experimental evaluation, and our method achieves much higher advantage than the existing ones on these more general OOD data sets (Colored MNIST and NICO). Moreover, NICO is a large-scale data set that covers a wide range of complicated correlation shift and diversity shift. To the best of our knowledge, it is one of the largest data sets that have been used in the field of theoretical investigations for IRM.
>
> We sincerely hope that all these new theoretical and experimental results can help revealing more in-depth properties and functions of our method.

---

> ### Author Response · Authors · 2024-12-02
> **Thanks for the good suggestions.**
>
> 1.1 Yes, $s$ is integrated in the inner integral, along with the contour $f^{-1}(\gamma)$. Then $\gamma$ is integrated in the outer integral from $-\infty$ to $+\infty$, serving as the _height_ for the contour. The whole TV integral is an analog to the Lipschitz-continuous surface area for the function $f$. More concretely, it is like computing the lateral surface area of a cone. We would add a figure in the next version to demonstrate this point.
>
> 1.2 \& 1.3 Indeed, this $\leftarrow$ symbol and the concept of measure need to be further explained. We would elaborate these terms more concretely in the next version. In brief, the induced measure $w\leftarrow \rho$ indicates the probability function for different classes ($w$ being the classifier) under the environment inference operator $\rho$. In Minimax-TV-$\ell_1$, $\rho$ should be learned to strengthen the TV penalty with the help of auxiliary variables, which provides environment information for $w$.
>
> 1.4 \& 1.5 We would make these remarks and other explanations regarding the game theory and the adversarial learning more prominent in the front part of this paper.
>
> In summary, we would provide more intuitive and concrete interpretations for the theory of this paper to improve its readability. In the future, we would further investigate the diversity shift problem by addressing the expectation terms in Eqs. 15 \& 16.

---

### Official Review · Reviewer_kusC · 2024-11-03

**Soundness:** 3
**Presentation:** 3
**Contribution:** 3
**Rating:** 6
**Confidence:** 4

**Summary:**

This paper proposes to learn the penalty term $\lambda$ in IRM-TV using a neural network to improve the OOD generalization performance of IRM-TV. As $\lambda$ is associated with the feature extractor $\Phi$, which also requires learning, $\lambda$ and $\Phi$ are learned alternatively via minimax adversarial learning. Experiments on a simulated dataset and real-world datasets show that learning $\lambda$ can improve IRM-TV's performance when distribution shifts.

**Strengths:**

1. Developing methods to learn $\lambda$ is important to improve IRM, which is a commonly used algorithm in OOD generalization.
2. The proposed minimax adversarial learning is novel and looks like a promising method to learn $\lambda$ well.
2. The writing is clear to follow.

**Weaknesses:**

My concern mainly falls in the experimental results:
1. The experiment seems only to be performed once as no standard deviation of the performance is reported.
2. Adversarial learning could be hard to converge. Analysis like Figure 1 on real-world datasets could be given to facilitate the understanding of how capable OOD-TV is when facing larger $\Phi$ and more complex datasets.

**Questions:**

1. Is there any repetition of experiments presented in section 4.2 to 4.4?
2. How do $g(\Psi, \Phi)$, $h(\Psi, \Phi, \rho)$, $||\Phi^{(k+1)}-\Phi^{(k)}||_2$ and $||\Psi^{(k+1)}-\Psi^{(k)}||_2$ change in real-world datasets as training goes by? Are $g(\Psi, \Phi)$ and $h(\Psi, \Phi, \rho)$ able to converge and how long does it take to converge?

---

> ### Author Response · Authors · 2024-11-21
> **Thanks for the constructive comments.**
>
> Please refer to the manuscript we have substantially revised and improved.
>
> 1. We try repeating the experiments in some cases for several times, and find that the results are similar, especially for the gaps between OOD-TV-IRM and IRM-TV, thus we just show one result for one case. In this revision, we add two new experiments with different tasks from the previous ones to evaluate OOD-TV-IRM in more general scenarios. In both tasks, OOD-TV-IRM performs significantly better than IRM-TV, as shown in Table 4.
>
> 2. The reason why it is difficult for an adversarial learning to converge is that the loss function decreases and increases alternately along with the interchanging between the primal update and the dual update, respectively. Nevertheless, we develop a convergent scheme in Theorem 3, and demonstrate a learning process on the real-world Colored MNIST data set in Appendix Figure A1. It successfully converges with $600$ epochs for OOD-TV-Minimax-$\ell_1$ or with $300$ epochs for OOD-TV-Minimax-$\ell_2$.

---

> ### Comment · Reviewer_kusC · 2024-11-26
>
> Thank you for the response.
> 1. I suggest including the experiment results of different repetitions in the paper by, for example, showing the average and standard deviation of accuracy/mean square error.
> 2. Figure A1 facilitates the understanding of how the training of OOD-TV progresses in real-world datasets.
>
> Most of my concerns are addressed. I would like to maintain my original score.

---

> > ### Author Response · Authors · 2024-11-26
> > **Thanks for the reply.**
> >
> > Thanks for the suggestion. We would arrange a mean$\pm$std version of these results in the final version.
> >
> > We have added new experiments on NICO, shown in Section 4.7 and Table 5 in the latest revision. It is a challenging data set including both correlation shift and diversity shift. Results show that OOD-TV-Minimax (ours) outperforms Minimax-TV significantly, which indicates that our OOD generalization methodology for Minimax-TV is tractable and effective in complicated scenarios with both correlation shift and diversity shift.

---

> ### Author Response · Authors · 2024-11-30
> **Latest new experimental results on NICO Animal.**
>
> We have added the experiments on **NICO Animal**, the other superclass besides NICO Vehicle. The experimental settings are nearly the same as those of NICO Vehicle. The accuracy results of different methods are: ZIN: 0.7596, OOD-TV-Minimax-$\ell_2$: **0.8528**, Minimax-TV-$\ell_1$: 0.7891, OOD-TV-Minimax-$\ell_1$: **0.8915**. Hence OOD-TV-Minimax (ours) outperforms Minimax-TV significantly, which indicates that our OOD generalization methodology for Minimax-TV is tractable and effective. All the required data sets have been used for experimental evaluation, and our method achieves much higher advantage than the existing ones on these more general OOD data sets (Colored MNIST and NICO). Moreover, NICO is a large-scale data set that covers a wide range of complicated correlation shift and diversity shift. To the best of our knowledge, it is one of the largest data sets that have been used in the field of theoretical investigations for IRM.
>
> We sincerely hope that all these new theoretical and experimental results can help revealing more in-depth properties and functions of our method.

---

> ### Comment · Reviewer_wMCo · 2024-12-02
>
> Thank you very much for the additional experiment results. The method has demonstrated some improvements over previous IRM methods. However, it did not compare with previous strong baselines on this kind of shifts.
>
> Another major concern (maybe the biggest) I would agree with the reviewer Hkpi  is that the math in the paper is really hard to follow: (On the other hand, it is evident that the value of a work shall not be judged based on the citations only, there are many first works ignored by the community because of lack of PR resources especially in the field of AI).
>
> 1.1 Take the definition for the total variation for example, In Eq (5), where is $s$ in the formula to be integrated? The conclusion that "this formulation shows the TV integrates allover all the contours of the function, reinforcing its capability in capturing piecewise-constant features" is a bit confusing here. It is suggested to add figures in the appendix to illustrate it.
>
> 1.2 In Eq (8), the usage of the symbol "<-" creates some burdens for people to quickly understand it. This symbol has many meanings. It could be approximating as time step proceeds or giving values to. People maybe able to read it by reading more texts in the main paragraph. But it indeed creates much unnecessary burdens which may not be hard to be fixed.
>
> 1.3 The same applies to line 162. Maybe using the term "measure" is more rigorous, but it does not help the readers to quickly get what the method is doing, and how it is different from previous methods used.
>
> 1.4 In Line 234, the paper tries to explain why the method works from the perspective of game theory, which is more concrete than the abstract and introduction parts. The authors can consider majorly revised the paper to improve the readability.
>
> 1.5 There are many examples like this.
>
> However, the above issue may not deny the technical contributions of this paper. This paper is suggested to make improvements in writing for clarity.

---

### Comment · Area_Chair_rrx7 · 2024-12-01

Hi Reviewers,

The authors have provided new results and responses - do have a look and engage with them in a discussion to clarify any remaining issues as the discussion period is coming to a close in less than a day (2nd Dec AoE).

Thanks for your service to ICLR 2025.

Best,
AC

---

### Meta-Review · Area_Chair_rrx7 · 2024-12-19

**Metareview:**

This work extends the previously proposed IRM-TV framework with a theoretically motivated algorithm for learning the TV penalty hyperparameter. All reviewers appreciated the technical contribution of the work, having both a novel algorithm and theoretical arguments to motivate its design. There were initial concerns about the lack of insight into the learning algorithm, and the lack of sufficient experiments to justify the effectiveness of the method. These issues were largely resolved by the comprehensive revision and rebuttal provided by the authors, that included results on several additional datasets as well as a new framing of the algorithm as a primal-dual method with additional convergence results. Another significant concern shared across reviewers was regarding the presentation of the work, specifically the lack of intuition for theoretical results and hard to read math, which were also largely addressed through the rebuttal, although reviewers still had remaining concerns on the clarity of the work (e.g. reviewer wMCo). Overall, the paper remained borderline following the discussion although reviewers acknowledged the technical contribution of the work, with 2 reviewers voting for acceptance and 1 non-responsive reviewer. The key remaining concerns are regarding presentation and lack of comparison with non-IRM baselines.

Overall, the AC agrees that this work makes a good contribution to the IRM approach to addressing generalization, with convincing improvements over related IRM baselines and theoretical results to motivate and justify the proposed method, and recommends acceptance to spur further development of this line of work. The AC agrees with reviewer wMCo that the significance of the work cannot be solely judged by citations as suggested by reviewer Hkpi, and it may take some time for promising approaches to mature. That being said, the AC also agrees with reviewers that the math is dense and more intuition should be provided to make the work more accessible, and that a comparison with other non-IRM methods will provide needed perspective on the merits of this line of work. The authors are encouraged to take these points into account when preparing the final version of the paper.

**Additional Comments On Reviewer Discussion:**

As mentioned above, authors included new experimental results on several datasets and a new theoretical framework to explain their learning algorithm and convergence properties in response to initial reviews. Reviewers were largely satisfied by the additional experiments and revision of the text of the paper (to provide needed intuition and insight instead of derviations), though there were remaining concerns that the math was still hard to read.

Authors did not include additional comparisons to other non IRM-style methods, which I thought was fine for now as the IRM approach is still maturing in comparison to other approaches.

How these points were weighed are included in the meta-review above.

---

### Decision · Program_Chairs · 2025-01-22

Accept (Poster)